# TurboBoA: Faster and Exact Attention-aware Quantization without Backpropagation

**Junhan Kim, Yeo Jeong Park, Seungwoo Son, Chungman Lee,**
**Ho-young Kim**, **Joonyoung Kim**, **Yongkweon Jeon**[*]
Samsung Research, Seoul, Korea
`{jun_one.kim, yeo_j.park, dragwon.jeon}@samsung.com`

## Abstract

The rapid growth of large language models (LLMs) has heightened the importance of post-training quantization (PTQ) for reducing memory and computation costs. Among PTQ methods, GPTQ has gained significant attention for its efficiency, enabling billion-scale LLMs to be quantized within a few GPU hours. However, GPTQ's assumption of layer-wise independence leads to severe accuracy drops in low-bit regimes. Recently, BoA improved upon GPTQ by incorporating inter-layer dependencies within attention modules, but its reliance on sequential quantization across all out-channels makes it substantially less efficient. In this paper, we propose TurboBoA, a new backpropagation-free PTQ algorithm that preserves the accuracy benefits of BoA while significantly accelerating the process. The proposed TurboBoA introduces three key innovations: (i) joint quantization of multiple out-channels with a closed-form error compensation rule, which reduces sequential bottlenecks and yields more than a three-fold speedup; (ii) a correction mechanism for errors propagated from preceding quantized layers; and (iii) adaptive grid computation with coordinate descent refinement to maintain alignment during iterative updates. Extensive experiments demonstrate that TurboBoA delivers substantial acceleration over BoA while consistently improving accuracy. When combined with outlier suppression techniques, it achieves state-of-the-art results in both weight-only and weight-activation quantization. The code will be available at `https://github.com/SamsungLabs/TurboBoA`.

## 1 Introduction

The rapid scaling of large language models (LLMs) (Touvron et al., 2023a;b) has dramatically increased their memory footprint and computational requirements, making deployment on resource-constrained hardware challenging. As a practical solution to reduce memory usage and accelerate inference, post-training quantization (PTQ), which reduces the precision of weights and activations using only a small calibration dataset, has received considerable attention.

The PTQ pipeline for LLMs typically involves two major stages. First, the model is transformed to be more robust to quantization by suppressing outliers in weights and activations through scaling (*e.g.*, SmoothQuant (Xiao et al., 2023)) or rotation (*e.g.*, QuaRot (Ashkboos et al., 2024)). Next, the transformed model is quantized under specific bit-width constraints. For weight quantization, backpropagation-free methods exploiting Hessian-guided error compensation have been widely adopted (Frantar et al., 2023; Kim et al., 2025; Li et al., 2025), as they facilitate efficient optimization of quantized weights without gradient-based training.

Among backpropagation-free methods, GPTQ is a representative approach known for its efficiency, enabling the quantization of billion-scale LLMs within a few GPU hours (Frantar et al., 2023). However, GPTQ assumes layer-wise independence, which leads to severe accuracy degradation in low-bit regimes (*e.g.*, INT2). Recently, BoA addressed this by exploiting attention reconstruction errors in the Hessian approximation (Kim et al., 2025). By capturing cross-layer dependencies within attention modules, BoA yields substantial accuracy gains over GPTQ. However, BoA introduces a

---

[*]Corresponding Author

significant computational bottleneck: it performs quantization *sequentially* across out-channels to compensate for the quantization error of each out-channel (see Fig. 1). Such sequential process, although necessary for precise error compensation, severely slows down the overall process and makes BOA substantially less efficient than GPTQ.

The primary goal of this paper is to accelerate BOA without sacrificing accuracy and even to achieve further performance improvements. Our main contributions are as follows:

- We propose TURBOBOA, which significantly accelerates BOA (**Section 3.1**). Our key idea is to quantize multiple out-channels simultaneously, thereby reducing the number of sequential operations while explicitly incorporating their dependencies into the error compensation (**Proposition 3.1**). Our timing measurements demonstrate that the proposed joint quantization leads to more than a three-fold speedup over BOA (**Table 2**).

- We incorporate two features into TURBOBOA to enhance its performance (**Sections 3.2 and 3.3**). First, TURBOBOA compensates for errors propagated from preceding quantized layers, mitigating error accumulation across layer depths (**Proposition 3.2**). Second, TURBOBOA adaptively determines quantization grids to align them with weights iteratively updated for the error compensation and further refines grids to reduce attention reconstruction errors (**Proposition 3.3**).

- From extensive experiments, we demonstrate that TURBOBOA delivers substantial acceleration over BOA while achieving superior accuracy (**Table 3**). Furthermore, when integrated with outlier suppression techniques, TURBOBOA achieves state-of-the-art results for both weight-only and weight-activation quantization (**Tables 4 and 5**).

**Notations** We use lowercase letters to denote vectors (*e.g.*, $\mathbf{w}$) and uppercase letters for matrices (*e.g.*, $\mathbf{W}$). $w_i$ denotes the $i$-th element in $\mathbf{w}$, and $W_{i,j}$ is the $(i, j)$-th entry in $\mathbf{W}$. We denote the $i$-th row of $\mathbf{W}$, which corresponds to the $i$-th out-channel, by $\mathbf{W}_{i,:}$ and the $j$-th column of $\mathbf{W}$ by $\mathbf{W}_{:,j}$. The submatrix of $\mathbf{W}$ consisting of the rows indexed by the index set $B$ is denoted by $\mathbf{W}_{B,:}$. Similarly, $\mathbf{W}_{:,B}$ denotes the submatrix of $\mathbf{W}$ with the columns indexed by $B$. $\mathbf{e}_i$ is the vector with a 1 in the $i$-th coordinate and 0's elsewhere, and $\mathbf{I}$ denotes the identity matrix. $\mathbf{0}_{d_1 \times d_2}$ and $\mathbf{1}_{d_1 \times d_2}$ are $(d_1 \times d_2)$-dimensional matrices with entries being zeros and ones, respectively.

## 2 RELATED WORKS

### 2.1 LLM QUANTIZATION

The main goal of PTQ is to minimize the degradation in task loss induced by quantization, which can be relaxed to the layer-wise reconstruction problem (LeCun et al., 1989; Nagel et al., 2020)

$$\min_{\mathbf{Q} \in \mathcal{Q}} \left\| (\mathbf{Q} - \mathbf{W}) \mathbf{X} \right\|_F^2, \tag{1}$$

where $\mathbf{W} \in \mathbb{R}^{d_{out} \times d_{in}}$ is a weight matrix for one layer, $\mathbf{X} \in \mathbb{R}^{d_{in} \times L}$ is its input of length $L$, and $\mathcal{Q}$ is the set of discrete quantized weights $\mathbf{Q}$. If channel-wise quantization is adopted, $\mathbf{Q}$ can be expressed as

$$\mathbf{Q} = \mathrm{diag}(\mathbf{s})\mathbf{W}_{int}, \mathbf{W}_{int} \in \{0, \ldots, 2^b - 1\}^{d_{out} \times d_{in}} \tag{2}$$

where $\mathbf{s} \in \mathbb{R}^{d_{out}}$ is a scale vector and $b$ is the target bit-width.

Early PTQ approaches aimed to reformulate the assignment of discrete quantized values into a continuous optimization problem, enabling quantized weights to be learned through gradient-based training. Representative algorithms include AdaRound, which introduced differentiable approximations for the rounding operation (Nagel et al., 2020), and BRECQ, which further extended this idea to the block-wise reconstruction problem to consider cross-layer dependencies (Li et al., 2021). Although these methods have been successful for small-scale models (*e.g.*, ResNet), they depend on time-consuming gradient-based training, which renders them impractical for LLMs with billions of parameters.

Recent research has therefore focused on developing cost-effective alternatives for LLM quantization (Frantar et al., 2023; Jeon et al., 2023; Kim et al., 2024; 2025). These works can be categorized

into two orthogonal classes. The first is backpropagation-free methods, which resort to Hessian-guided error compensation (*e.g.*, GPTQ (Frantar et al., 2023) and BOA (Kim et al., 2025)). The second is transformation-based methods, which suppress outliers via scaling or rotation, thereby transforming LLMs into a more quantization-friendly form (*e.g.*, SmoothQuant (Xiao et al., 2023) and QuaRot (Ashkboos et al., 2024)).

Our approach belongs to the backpropagation-free class and further improves BOA by enhancing both efficiency and accuracy. Furthermore, similar to GPTQ and BOA, our method can be effectively combined with transformation-based methods, demonstrating strong complementarity between the two classes.

## 2.2 BACKPROPAGATION-FREE WEIGHT QUANTIZATION

Backpropagation-free PTQ algorithms, which rely on the Hessian-guided error compensation, have been widely adopted for efficient LLM quantization (Frantar et al., 2023; Li et al., 2025; Kim et al., 2025). These algorithms rapidly quantize LLMs by iteratively conducting quantization and error correction, which is given as (Frantar et al., 2023)

$$\Delta\mathbf{w} = -\frac{w_p - q_p}{U_{p,p}}\mathbf{U}_{p,:} \text{ where } \mathbf{U} = \text{Chol}(\mathbf{H}^{-1})^T, \tag{3}$$

where $q_p$ is the quantized version of the weight $w_p$, $\mathbf{H}$ is the Hessian matrix, and $\text{Chol}(\cdot)$ denotes a Cholesky decomposition, that is, $\mathbf{U}$ is upper triangular such that $\mathbf{H}^{-1} = \mathbf{U}^T\mathbf{U}$.

The first algorithm that successfully scaled this principle to LLMs was GPTQ (Frantar et al., 2023). However, GPTQ approximates the Hessian based on layer-wise reconstruction errors, failing to account for inter-layer dependencies and resulting in suboptimal performance, particularly at low bit-widths (*e.g.*, INT2). Recently, BOA addressed this issue by exploiting attention reconstruction errors in the Hessian approximation (Kim et al., 2025). The resulting Hessians explicitly model dependencies between out-channels (see $\mathbf{H}_{out}$ in Table 1), enabling the error compensation for each out-channel and yielding substantial accuracy gains over GPTQ. Nevertheless, such improved Hessians make sequential processing across out-channels unavoidable; the second out-channel can be quantized after compensating for the quantization error induced by the first out-channel, which differs from GPTQ that quantizes all out-channels simultaneously (see Fig. 1). To alleviate this bottleneck, BOA parallelizes quantization across different attention heads (*e.g.*, quantizing the first out-channel of all heads concurrently) by assuming head-wise independence. While this strategy provides some acceleration, BOA still remains substantially more time-consuming than GPTQ, highlighting a trade-off between accuracy and efficiency.

Table 1: Loss used to approximate Hessians and the corresponding Hessians in GPTQ and BOA.

| Method | Layer | Loss ($\|\mathbf{G}\Delta\mathbf{W}\mathbf{X}\|_F^2$) | $\mathbf{H} = \mathbf{H}_{in} \otimes \mathbf{H}_{out}$ |
|---|---|---|---|
| GPTQ | $\mathbf{W}_{\{Q,K,V\}}$ | $\|\Delta\mathbf{W}\mathbf{X}\|_F^2$ | $\mathbf{X}\mathbf{X}^T \otimes \mathbf{I}$ |
| BOA | $\mathbf{W}_{Q,h}$ | $\|\mathbf{K}_h\Delta\mathbf{W}_{Q,h}\mathbf{X}\|_F^2$ | $\mathbf{X}\mathbf{X}^T \otimes \mathbf{K}_h^T\mathbf{K}_h$ |
| | $\mathbf{W}_{K,h}$ | $\|\mathbf{Q}_h\Delta\mathbf{W}_{K,h}\mathbf{X}\|_F^2$ | $\mathbf{X}\mathbf{X}^T \otimes \mathbf{Q}_h^T\mathbf{Q}_h$ |
| | $\mathbf{W}_{V,h}$ | $\|\mathbf{W}_{out,h}\Delta\mathbf{W}_{V,h}\mathbf{X}\mathbf{A}_h^T\|_F^2$ | $\mathbf{X}\mathbf{A}_h^T\mathbf{A}_h\mathbf{X}^T \otimes \mathbf{W}_{out,h}^T\mathbf{W}_{out,h}$ |

[*] $h$ denotes the index of the attention head.

## 3 TURBOBOA

We now introduce the proposed TURBOBOA. To enhance both the efficiency and accuracy of BOA, we introduce three key innovations, each of which will be described in the following subsections in detail.

## 3.1 SIMULTANEOUS QUANTIZATION OF MULTIPLE OUT-CHANNELS

As described earlier, BOA sequentially quantizes out-channels one by one. This means that when quantizing a weight matrix with 128 out-channels (*e.g.*, query, key, and value projection weights in Llama3-8B), BoA requires 128 sequential operations. Consequently, BoA is substantially more time-consuming than GPTQ, in which all out-channels are quantized in parallel.

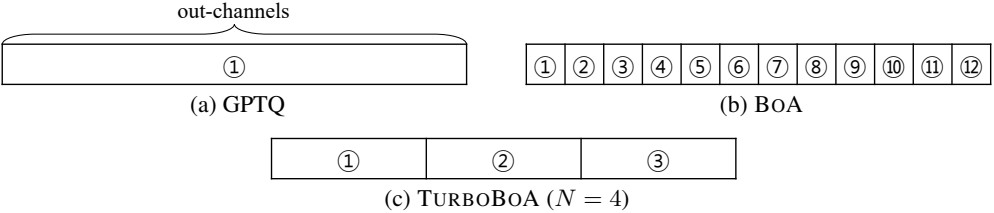

Figure 1: Quantization orders in GPTQ, BOA, and the proposed TURBOBOA. (a) GPTQ quantizes all out-channels jointly but without error correction. (b) BOA compensates for the quantization error but requires fully sequential processing across out-channels. (c) TURBOBOA reduces sequential operations by quantizing multiple $N$ out-channels jointly while still applying error compensation.

To accelerate the quantization process, TURBOBOA quantizes multiple $N$ out-channels simultaneously, thereby reducing the number of sequential operations (see Fig. 1). In the previous example, when $N = 16$, the number of sequential operations decreases from 128 to 8. We note that while $N$ out-channels are quantized together as if they were mutually independent (as in GPTQ), we explicitly incorporate their dependencies into the error compensation step. To do so, instead of naïvely adding weight compensation for each out-channel, we formulate the problem of compensating for the errors of multiple out-channels as

$$\min_{\Delta \mathbf{W}} \ \|\mathbf{G}\Delta\mathbf{W}\mathbf{X}\|_F^2,$$
$$\text{s.t.} \quad \mathbf{e}_i^T \Delta\mathbf{W} = \mathbf{Q}_{i,:} - \mathbf{W}_{i,:} \ (0 \le i < N), \tag{4}$$

where $\mathbf{Q}_{i,:}$ is the quantized version of $\mathbf{W}_{i,:}$ and we use the unified notation $\|\mathbf{G}\Delta\mathbf{W}\mathbf{X}\|_F^2$ to denote the attention reconstruction errors in Table 1 (*e.g.*, $\mathbf{G} = \mathbf{K}_h$ for the query projection weight $\mathbf{W}_{Q,h}$). In the following proposition, we present a closed-form solution to (4).

**Proposition 3.1.** *Let $\mathbf{W}$ be a matrix whose Hessian is given as $\mathbf{H} = \mathbf{H}_{in} \otimes \mathbf{H}_{out}$. Suppose the first $N$ out-channels of $\mathbf{W}$ have been quantized simultaneously and the other out-channels are updated to minimize the attention reconstruction error in (4). Then, the update $[\Delta\mathbf{W}]_{N:,:}$ satisfies*

$$[\Delta\mathbf{W}]_{N:,:} = -[\mathbf{U}_{out}^T]_{N:,B}[\mathbf{U}_{out}^T]_{B,B}^{-1}(\mathbf{W}_{B,:} - \mathbf{Q}_{B,:}), \tag{5}$$

*where $B = \{0, \ldots, N-1\}$ and $\mathbf{U}_{out} = \text{Chol}(\mathbf{H}_{out}^{-1})^T$.*

*Proof.* See Appendix C. ∎

A pertinent question is whether such joint quantization inevitably leads to accuracy degradation. In the conventional BOA, the quantization error of the first out-channel can be compensated by all subsequent out-channels; *e.g.*, 127 out-channels participate in the error compensation in the previous example. Whereas, in our approach, multiple out-channels are quantized at once, so the number of out-channels available for the error correction decreases (*e.g.*, 127→112 when $N = 16$). In Section 4.2, we will empirically show that the degradation arising from the reduced error correction flexibility is negligible, even in the low-bit regime (see Table 2).

### 3.2 ERROR COMPENSATION FOR PRE-QUANTIZED LAYERS

Another limitation of BOA is that it does not account for quantization errors propagated from previously quantized layers. During quantization, errors produced in one layer propagate to subsequent layers by perturbing their input distributions, and these deviations accumulate as the network depth increases, as reported in GPTAQ (Li et al., 2025).

Let $\widetilde{\mathbf{X}}$ be the original full-precision (FP) input. We observe that the input deviation $\Delta\mathbf{X} := \mathbf{X} - \widetilde{\mathbf{X}}$ introduces additional distortion in the attention output as follows:

$$\mathbf{G}\mathbf{Q}\mathbf{X} - \mathbf{G}\mathbf{W}\widetilde{\mathbf{X}} = \mathbf{G}(\mathbf{Q} - \mathbf{W})\mathbf{X} + \mathbf{G}\mathbf{W}(\mathbf{X} - \widetilde{\mathbf{X}}) = \mathbf{G}\Delta\mathbf{W}\mathbf{X} + \mathbf{G}\mathbf{W}\Delta\mathbf{X}. \tag{6}$$

Here, $\mathbf{G}\Delta\mathbf{W}\mathbf{X}$ corresponds to the error introduced by quantizing the current layer, while $\mathbf{G}\mathbf{W}\Delta\mathbf{X}$ captures the output deviation induced by the perturbed input. We explicitly incorporate the additional distortion $\mathbf{G}\mathbf{W}\Delta\mathbf{X}$ into the error compensation. Specifically, after quantizing $N$ out-channels

---

**Algorithm 1** TURBOBOA

---

**Input**: weights $\mathbf{W}_{\{Q,K,V\}} \in \mathbb{R}^{H \times d_h \times d}$, target bit-width $b$, inputs $\mathbf{X} \in \mathbb{R}^{d \times L}$, FP representation $\widetilde{\mathbf{X}} \in \mathbb{R}^{d \times L}$, number $N$ of out-channels quantized simultaneously, and stabilization coefficient $\alpha$

**Output**: quantized weights $\mathbf{Q}_{\{Q,K,V\}}$
1: **for** $\mathbf{W} \in \{\mathbf{W}_Q, \mathbf{W}_K, \mathbf{W}_V\}$ **do**
2:    Initialize quantized outputs and integer weights: $\mathbf{Q}_h, \mathbf{W}_{int,h} \leftarrow \mathbf{0}_{d_h \times d}$
3:    Initialize out-channel scales: $\mathbf{s}_h \leftarrow \mathbf{1}_{d_h \times 1}$
4:    Compute attention-aware Hessians $\mathbf{H}_{in,h}$ and $\mathbf{H}_{out,h}$
5:    Compute $\mathbf{U}_{in,h} = \mathrm{Chol}(\mathbf{H}_{in,h}^{-1})^T$, $\mathbf{U}_{out,h} = \mathrm{Chol}(\mathbf{H}_{out,h}^{-1})^T$, and $\mathbf{R} = \alpha(\mathbf{X} - \widetilde{\mathbf{X}})\mathbf{X}^T$
6:    Initialize updated weights: $\widetilde{\mathbf{W}} \leftarrow \mathbf{W}$
7:    **for** $i = 0, N, 2N, \ldots$ **do**
8:       Take $N$ out-channels to be quantized jointly: $\mathbf{W}^{(i)} \leftarrow [\widetilde{\mathbf{W}}_h]_{B,:}$ ($B = \{i, i+1, \ldots, i+N-1\}$)
9:       Set scales: $[\mathbf{s}_h]_B \leftarrow \min_{\mathbf{s}} \mathrm{tr}(\Delta\mathbf{W}^{(i)}\mathbf{H}_{in,h}(\Delta\mathbf{W}^{(i)})^T)$
10:      Quantize $\mathbf{W}^{(i)}$: $([\mathbf{Q}_h]_{B,:}, [\mathbf{W}_{int,h}]_{B,:}) \leftarrow \mathrm{GPTAQ}(\mathbf{W}^{(i)}, \mathbf{U}_{in,h}, \mathbf{R}, [\mathbf{s}_h]_B)$     ▷ see Algorithm 3
11:      Update remaining out-channels:

$$[\widetilde{\mathbf{W}}_h]_{i+N:,:} \leftarrow [\widetilde{\mathbf{W}}_h]_{i+N:,:} - [\mathbf{U}_{out,h}^T]_{i+N:,B}[\mathbf{U}_{out,h}^T]_{B,B}^{-1}([\widetilde{\mathbf{W}}_h]_{B,:} - [\mathbf{Q}_h]_{B,:})$$
$$+ [\mathbf{U}_{out,h}^T]_{i+N:,B}[\mathbf{U}_{out,h}^T]_{B,B}^{-1}[\widetilde{\mathbf{W}}_h]_{B,:}\mathbf{R}\mathbf{H}_{in,h}^{-1}$$

12:    Refine scales: $\mathbf{s}_h \leftarrow \min_{\mathbf{s}} \|\mathbf{G}_h(\mathrm{diag}(\mathbf{s})\mathbf{W}_{int,h} - \mathbf{W}_h)\mathbf{X} + \mathbf{G}_h\mathbf{W}_h\Delta\mathbf{X}\|_F^2$    ▷ see Algorithm 2
13:    Update quantized weights: $\mathbf{Q}_h \leftarrow \mathrm{diag}(\mathbf{s}_h)\mathbf{W}_{int,h}$

---

$\mathbf{W}_{B,:}$, we compensate for both the error introduced by the weight perturbation (*i.e.*, $\mathbf{G}_{:,B}\Delta\mathbf{W}_{B,:}\mathbf{X}$) and the error incurred by the input deviation $\Delta\mathbf{X}$ (*i.e.*, $\mathbf{G}_{:,B}\mathbf{W}_{B,:}\Delta\mathbf{X}$), which reformulates the error compensation problem in (4) as

$$\min_{\Delta\mathbf{W}} \|\mathbf{G}\Delta\mathbf{W}\mathbf{X} + \mathbf{G}_{:,B}\mathbf{W}_{B,:}\Delta\mathbf{X}\|_F^2,$$
$$\text{s.t.} \quad \mathbf{e}_i^T\Delta\mathbf{W} = \mathbf{Q}_{i,:} - \mathbf{W}_{i,:} \ (0 \le i < N). \tag{7}$$

The following proposition provides a closed-form solution to the above problem.

**Proposition 3.2.** *Let $\mathbf{W}$ be a matrix whose Hessian is given as $\mathbf{H} = \mathbf{H}_{in} \otimes \mathbf{H}_{out}$. Suppose the first $N$ out-channels of $\mathbf{W}$ have been quantized simultaneously, where the input $\mathbf{X}$ is distorted from the FP representation $\widetilde{\mathbf{X}}$ due to the quantization errors produced in earlier layers. Then, the update $[\Delta\mathbf{W}]_{N:,:}$ of the other out-channels to minimize the attention reconstruction error in (7) is*

$$[\Delta\mathbf{W}]_{N:,:} = -[\mathbf{U}_{out}^T]_{N:,B}[\mathbf{U}_{out}^T]_{B,B}^{-1}\left((\mathbf{W}_{B,:} - \mathbf{Q}_{B,:}) - \mathbf{W}_{B,:}\mathbf{R}\mathbf{H}_{in}^{-1}\right), \tag{8}$$

*where $B = \{0, \ldots, N-1\}$, $\mathbf{U}_{out} = \mathrm{Chol}(\mathbf{H}_{out}^{-1})^T$, and $\mathbf{R} = \Delta\mathbf{X}\mathbf{X}^T$.*

*Proof.* See Appendix D.      ■

Compared to the update rule in (5) for the first layer, the update rule in (8) involves the additional term related to the input deviation $\Delta\mathbf{X}$, which explicitly compensates for errors propagated across quantized layers and ensures the quantized model to replicate the behavior of the FP model more faithfully across all layers.

We note that our approach of incorporating the input deviation $\Delta\mathbf{X}$ is motivated by the error compensation framework of GPTAQ (Li et al., 2025). However, a key technical distinction lies in the structure of the Hessian $\mathbf{H}_{out}$. While GPTAQ assumes $\mathbf{H}_{out} = \mathbf{I}$, which decouples out-channels and simplifies the optimization into a set of independent vector equations, our framework addresses a general (potentially dense) $\mathbf{H}_{out}$ to incorporate dependencies within out-channels. This transition from a separable vector-wise optimization to a coupled matrix-wise formulation requires a more sophisticated derivation of the closed-form update rule, as detailed in Appendix D.

### 3.3 ADAPTIVE GRID SELECTION WITH ATTENTION-WISE REFINEMENT

A remaining limitation of BOA lies in its grid computation. First, once initialized, the quantization grid remains fixed throughout the iterative process (Kim et al., 2025). However, since out-channels

are continuously updated due to the error compensation, the initial grid becomes misaligned with the updated weights. This misalignment would particularly be severe in low-bit regimes, where large weight perturbations $(\mathbf{W}_{B,:} - \mathbf{Q}_{B,:})$ result in large updates $[\Delta \mathbf{W}]_{N:,:}$ (see (5)). Second, BOA computes the grid in a way to minimize the layer-wise reconstruction loss, which is not aligned with the goal of minimizing the attention reconstruction error.

To address these issues, TURBOBOA determines the quantization grid immediately before each out-channel is quantized (line 9 in Algorithm 1), which ensures that every quantization step uses a grid aligned to the previously updated weights. To reduce unnecessary overhead, the grid is computed exclusively for the out-channels to be quantized at each quantization step. Furthermore, we introduce a grid refinement step (line 12 in Algorithm 1). At this stage, we *freeze* the integer weights $\mathbf{W}_{int} \in \{0, 1, \ldots, 2^b - 1\}^{d_{out} \times d_{in}}$ assigned through the iterative process (lines 7-11 in Algorithm 1) and refine only scales $\mathbf{s} \in \mathbb{R}^{d_{out}}$ to further reduce the attention reconstruction error in (6):

$$\min_{\mathbf{s}} \|\mathbf{G}(\text{diag}(\mathbf{s})\mathbf{W}_{int} - \mathbf{W})\mathbf{X} + \mathbf{G}\mathbf{W}\Delta\mathbf{X}\|_F^2. \tag{9}$$

To solve this problem, we adopt coordinate descent (CD), which iteratively updates one scale at a time while fixing the others. The following proposition presents the closed-form update rule for each CD step, which facilitates each scale update without a costly numerical optimization.

**Proposition 3.3.** *Let $\mathbf{W}$ be a matrix whose Hessian is given as $\mathbf{H} = \mathbf{H}_{in} \otimes \mathbf{H}_{out}$. Suppose $\mathbf{W}$ has been quantized to $\mathbf{Q} = \text{diag}(\mathbf{s})\mathbf{W}_{int}$ where $\mathbf{s}$ is the scale vector and $\mathbf{W}_{int}$ is the fixed integer weights. Suppose the scales $\mathbf{s}$ are refined to minimize the attention reconstruction error in (9) via CD. Then, the update rule for each CD step is given as*

$$s_j^* = s_j + \frac{[\mathbf{W}_{int}(\mathbf{H}_{in}(\mathbf{W} - \mathbf{Q})^T - \mathbf{R}^T\mathbf{W}^T)\mathbf{H}_{out}]_{j,j}}{[\mathbf{W}_{int}\mathbf{H}_{in}\mathbf{W}_{int}^T]_{j,j}[\mathbf{H}_{out}]_{j,j}},$$

*where $\mathbf{R} = \Delta\mathbf{X}\mathbf{X}^T$.*

*Proof.* See Appendix E. ∎

After refining the scales, we update the quantized weights (line 13 in Algorithm 1), which yields the final output of the proposed TURBOBOA.

## 4 EXPERIMENTS

### 4.1 EXPERIMENTAL SETUP

We evaluate the performance of TURBOBOA using Llama models (Touvron et al., 2023a;b). Following prior works (Ashkboos et al., 2024; Liu et al., 2024; Kim et al., 2025), we use 128 sequences of length 2048 randomly sampled from the WikiText-2 (Wiki2) dataset Merity et al. (2016) as calibration data for quantization. As a performance metric, we use perplexity (PPL) on the Wiki2 and C4 (Raffel et al., 2020) test sets and the average accuracy across eight zero-shot commonsense reasoning tasks.[1] All experiments were conducted using NVIDIA H100 GPUs (80 GB). While a single GPU was sufficient for most models, we utilized two GPUs for the 70B model to accommodate its larger memory requirements.

**Hessian** We adopted Hessians derived in BOA (see Table 1), since they are currently the most accurate closed-form Hessians available in the literature. However, we emphasize that our main results in Propositions 3.1-3.3 are not specific to BOA's Hessians and can be applied to any Kronecker-structured Hessians $\mathbf{H} = \mathbf{H}_{in} \otimes \mathbf{H}_{out}$. Consequently, our method can directly leverage more advanced Hessian formulations once they become available.

---

[1]ARC-challenge/easy (Clark et al., 2018), BoolQ (Clark et al., 2019), HellaSwag (Zellers et al., 2019), LAMBADA (Paperno et al., 2016), OpenbookQA (Mihaylov et al., 2018), PIQA (Bisk et al., 2020), and Wino-Grande (Sakaguchi et al., 2021)

Table 2: Ablation of multiple-row processing (INT2 quantization)

| Method | $N$ | Llama3.2-1B | | Llama3.2-3B | | Llama3-8B | | Llama3.1-70B | |
|---|---|---|---|---|---|---|---|---|---|
| | | Time (min) | Wiki2 ($\downarrow$) | Time (min) | Wiki2 ($\downarrow$) | Time (min) | Wiki2 ($\downarrow$) | Time (hr) | Wiki2 ($\downarrow$) |
| BoA | 1 | 13.32 | 40.40 | 59.94 | 32.26 | 94.75 | 15.20 | 16.99 | 7.726 |
| BoA + **F1** | 4 | **6.255** | 41.09 | **22.68** | 32.21 | **39.46** | 15.27 | **7.683** | 7.721 |
| | 8 | **5.002** | 41.53 | **16.01** | 31.66 | **30.55** | 15.30 | **6.274** | 7.714 |
| | 16 | **4.363** | 41.85 | **12.70** | 31.99 | **25.30** | 15.41 | **5.636** | 7.758 |
| | 32 | **3.985** | 41.75 | **11.01** | 32.15 | **22.95** | 15.22 | **5.060** | 7.746 |
| | 64 | - | - | **10.29** | 32.31 | **21.56** | 15.44 | **4.885** | 7.774 |

[*] Following BoA (Kim et al., 2025), QuaRot has been applied before quantizing weights. We note that TurboBoA reduces to GPTQ under $N = 64$ for Llama3.2-1B and $N = 128$ for other models.

Table 3: Ablation of features targeting performance enhancement (INT2 quantization)

| Method | F2 | F3 | Llama3.2-1B | | | Llama3.2-3B | | | Llama3-8B | | |
|---|---|---|---|---|---|---|---|---|---|---|---|
| | | | Wiki2 ($\downarrow$) | C4 ($\downarrow$) | Time | Wiki2 ($\downarrow$) | C4 ($\downarrow$) | Time | Wiki2 ($\downarrow$) | C4 ($\downarrow$) | Time |
| BoA | | | 40.40 | 104.9 | 13.32 | 32.26 | 79.17 | 59.94 | 15.20 | 36.95 | 94.75 |
| TurboBoA ($N = 16$) | | | 41.85 | 108.1 | 4.363 | 31.99 | 80.09 | 12.70 | 15.41 | 38.96 | 25.30 |
| | ✓ | | 37.15 | 92.58 | 6.253 | 25.92 | 63.48 | 17.51 | 14.21 | 34.67 | 40.16 |
| | | ✓ | 39.45 | 107.3 | 4.426 | 31.12 | 73.57 | 12.84 | 15.01 | 36.40 | 25.39 |
| | ✓ | ✓ | 33.33 | 85.55 | 6.263 | 24.10 | 54.20 | 17.71 | 13.54 | 32.99 | 40.20 |

[*] Time in minutes. QuaRot has been applied before quantizing weights.

**Joint Quantization Hyperparameter** $N$    Our ablation study on the number $N$ of jointly quantized out-channels indicates that significant speedups are achievable up to $N = 16$, beyond which further increases (*e.g.*, $N = 32$ or $64$) yield only marginal gains. To ensure stability, we conservatively set $N = 16$ for all main experiments. A detailed analysis is provided in Section 4.2.

**CD-based Scale Refinement**    We set the number of CD iterations to one (*i.e.*, $n_{iter} = 1$ in Algorithm 2; see Appendix E), as additional iterations yield only marginal improvements. The corresponding ablation study can be found in Appendix F.4.

**Stabilization Coefficient** $\alpha$    Following the implementation of GPTAQ (Li et al., 2025), we introduce a stabilization coefficient $\alpha$ to modulate the impact of the input deviation $\Delta \mathbf{X}$ arising from the quantization errors of preceding layers (see line 5 in Algorithm 1). This coefficient acts as a regularization parameter that prevents the compensation term from over-adjusting to accumulated distortions, which could otherwise lead to numerical instability. In our experiments, we evaluated $\alpha \in \{0.05, 0.125, 0.25\}$ and reported the best-performing result for each model.

## 4.2    Ablation Studies

Recall that we incorporated three key features into the conventional BoA to accelerate the overall process and enhance the quantization performance. To validate the effectiveness of each feature, we conduct ablation studies.

**Speedup**    We first investigate the efficacy of the joint quantization of multiple $N$ out-channels (**F1**), introduced to mitigate the sequential bottleneck of BoA. Specifically, we measure the processing time by varying $N \in \{4, 8, 16, 32, 64\}$. As expected, the processing time decreases significantly as more out-channels are quantized simultaneously (see Table 2); for example, when $N = 16$, TurboBoA achieves more than a three-fold speedup. In particular, for the 70B model, this translates to a saving of 9~12 hours in absolute terms, demonstrating a substantial gain that becomes more impactful as model scale increases.

Intuitively, increasing $N$ reduces the degrees of freedom available for error compensation. However, our empirical results reveal that performance degradation remains negligible up to $N = 64$, suggesting that the remaining out-channels provide sufficient capacity and the proposed update rule in (5) effectively captures inter-channel correlations to compensate for joint quantization errors. We leave a formal theoretical characterization of the error bounds with respect to $N$ as an interesting open question. While this robustness to a large $N$ allows for aggressive parallelization, we observe that the speedup gain diminishes beyond $N = 16$. Therefore, we conservatively set $N = 16$ for the remaining experiments to retain a higher margin of flexibility for error compensation.

Table 4: Weight-only quantization performance on transformed Llama2 and Llama3 models

(a) PPL ($\downarrow$)

| Precision | Transform | Quantizer | Llama3.2-1B | | Llama3.2-3B | | Llama3-8B | | Llama2-7B | | Llama2-13B | |
|---|---|---|---|---|---|---|---|---|---|---|---|---|
| | | | Wiki2 | C4 | Wiki2 | C4 | Wiki2 | C4 | Wiki2 | C4 | Wiki2 | C4 |
| FP16 | Baseline | | 13.16 | 21.31 | 11.05 | 16.49 | 6.139 | 9.444 | 5.473 | 7.266 | 4.885 | 6.730 |
| INT3 | OmniQuant† | RTN | - | - | - | - | - | - | 6.640 | 9.383 | 5.593 | 7.989 |
| | DuQuant | RTN | 2.7e4 | 1.8e4 | 15.18 | 22.31 | 10.78 | 17.90 | 6.226 | 8.645 | 5.414 | 7.598 |
| | SpinQuant | RTN | 18.04 | 31.06 | 12.29 | 21.79 | 8.352 | 14.55 | 6.456 | 10.11 | 5.576 | 8.595 |
| | | GPTQ | 16.21 | 27.60 | 12.87 | 20.47 | 7.438 | 12.75 | 6.001 | 8.619 | 5.299 | 7.682 |
| | QuaRot | RTN | 98.24 | 139.0 | 89.54 | 101.1 | 38.64 | 51.43 | 129.2 | 111.9 | 48.06 | 48.79 |
| | | GPTQ | 16.56 | 27.28 | 13.58 | 20.48 | 7.490 | 12.92 | 6.122 | 8.688 | 5.382 | 7.706 |
| | | BoA | 15.73 | 26.15 | 12.97 | 19.96 | 7.145 | 12.25 | 5.874 | 8.268 | 5.202 | 7.436 |
| | | **TurboBoA** | **15.49** | **26.09** | **12.54** | **19.43** | **7.116** | **12.23** | **5.850** | **8.248** | **5.185** | **7.422** |
| INT2 | OmniQuant† | RTN | - | - | - | - | - | - | 21.85 | 39.34 | 12.92 | 19.99 |
| | DuQuant | RTN | 9.3e3 | 1.6e4 | 770.9 | 905.7 | 2.6e4 | 1.8e5 | 46.27 | 69.02 | 10.40 | 15.35 |
| | SpinQuant | RTN | 68.80 | 144.1 | 33.91 | 73.09 | 21.52 | 44.30 | 16.95 | 29.21 | 9.742 | 16.25 |
| | | GPTQ | 48.64 | 127.1 | 34.65 | 92.42 | 15.86 | 39.11 | 15.43 | 30.30 | 9.652 | 19.35 |
| | QuaRot | RTN | 2.6e5 | 2.5e5 | 2.3e4 | 1.1e4 | 3.5e5 | 3.6e5 | 1.1e4 | 1.1e4 | 7.9e3 | 6.2e3 |
| | | GPTQ | 54.28 | 118.6 | 52.18 | 128.8 | 18.28 | 48.31 | 22.05 | 41.92 | 9.593 | 19.47 |
| | | BoA | 40.86 | 107.9 | 33.40 | 79.21 | 15.24 | 36.82 | 10.42 | 19.17 | 8.237 | 14.66 |
| | | **TurboBoA** | **33.33** | **85.55** | **24.10** | **54.20** | **13.54** | **32.99** | **9.108** | **16.64** | **7.337** | **13.04** |

(b) Zero-shot Accuracy ($\uparrow$)

| Precision | Transform | Quantizer | Llama3.2-1B | Llama3.2-3B | Llama3-8B | Llama2-7B | Llama2-13B |
|---|---|---|---|---|---|---|---|
| FP16 | Baseline | | 56.82 | 63.01 | 70.34 | 67.28 | 69.83 |
| INT3 | OmniQuant† | RTN | - | - | - | 60.25 | 65.44 |
| | DuQuant | RTN | 31.12 | 54.59 | 52.42 | 63.24 | 67.04 |
| | SpinQuant | RTN | 48.65 | 56.69 | 64.32 | 60.40 | 65.32 |
| | | GPTQ | 51.33 | 59.39 | 67.05 | 64.34 | 67.62 |
| | QuaRot | RTN | 38.05 | 35.99 | 42.80 | 31.71 | 36.86 |
| | | GPTQ | 51.13 | 57.89 | 66.67 | 63.72 | 67.79 |
| | | BoA | 52.46 | 60.31 | 68.09 | 64.44 | 68.55 |
| | | **TurboBoA** | **53.32** | **61.26** | **68.57** | **65.21** | **69.07** |
| INT2 | OmniQuant† | RTN | - | - | - | 37.92 | 44.14 |
| | DuQuant | RTN | 30.42 | 30.56 | 30.69 | 32.30 | 45.85 |
| | SpinQuant | RTN | 35.97 | 37.94 | 42.25 | 38.95 | 47.41 |
| | | GPTQ | 36.50 | 39.71 | 46.78 | 43.03 | 49.50 |
| | QuaRot | RTN | 31.04 | 31.85 | 30.71 | 30.27 | 29.91 |
| | | GPTQ | 36.43 | 39.17 | 45.02 | 38.98 | 49.51 |
| | | BoA | 38.67 | 43.86 | 50.29 | 51.00 | 56.92 |
| | | **TurboBoA** | **40.31** | **45.85** | **52.59** | **53.27** | **59.69** |

† The official code does not support models exploiting grouped query attention.

**Performance Enhancement** We next examine the effectiveness of two features introduced to enhance the performance of TurboBoA: error compensation for pre-quantized layers (**F2**) and adaptive grid computation with CD-based refinement (**F3**). As shown in Table 3, incorporating either feature individually leads to consistent improvements. For example, on Llama3.2-1B, **F2** reduces PPL from 41.85 to 37.15 on Wiki2 and from 108.1 to 92.58 on C4, highlighting the benefit of mitigating error accumulation across layer depths. Similarly, **F3** improves alignment with the updated weight distribution, lowering PPL to 39.45 and 107.3 on Wiki2 and C4, respectively. Notably, the combination of both features yields the best performance, demonstrating their complementary roles; TurboBoA achieves PPLs of 33.33 on Wiki2 and 85.55 on C4 for Llama3.2-1B, representing substantial reductions over the baseline BoA. Consistent trends are observed in larger models, confirming that these enhancements generalize effectively across scales.

Finally, we analyze the runtime overhead introduced by these features. While **F3** adds only a marginal cost (*e.g.*, approximately one minute for Llama3-8B), the overhead of **F2** is more noticeable as it requires an additional forward pass of the FP model to compute the input deviation $\Delta \mathbf{X}$. However, we emphasize that this is a fixed, one-time cost, as the FP activation $\tilde{\mathbf{X}}$ is independent of the quantization process. Despite this overhead, TurboBoA still completes the entire quantization process substantially faster than BoA, which confirms that the efficiency gains from reducing sequential operations via **F1** more than compensate for the additional computations required for accuracy enhancement.

Table 5: Weight-activation quantization performance on transformed Llama2 and Llama3 models

(a) PPL (↓)

| Precision | Transform | Quantizer | Llama3.2-1B | | Llama3.2-3B | | Llama3-8B | | Llama2-7B | | Llama2-13B | |
|---|---|---|---|---|---|---|---|---|---|---|---|---|
| | | | Wiki2 | C4 | Wiki2 | C4 | Wiki2 | C4 | Wiki2 | C4 | Wiki2 | C4 |
| FP16 | Baseline | | 13.16 | 21.31 | 11.05 | 16.49 | 6.139 | 9.444 | 5.473 | 7.266 | 4.885 | 6.730 |
| W2A4KV16 | OmniQuant† | RTN | - | - | - | - | - | - | 2.3e3 | 3.2e3 | 2.8e3 | 4.4e3 |
| | DuQuant | RTN | 1.0e4 | 1.5e4 | 1.2e3 | 1.6e3 | 4.4e4 | 2.5e5 | 375.0 | 514.3 | 13.25 | 20.12 |
| | SpinQuant | GPTQ | 104.4 | 235.3 | 68.74 | 173.7 | 26.35 | 76.71 | 24.19 | 49.21 | 13.61 | 28.30 |
| | | BOA | 59.95 | 136.5 | 34.24 | 110.0 | 17.31 | 48.04 | 11.27 | 19.86 | 8.652 | 15.33 |
| | | TURBOBOA | 49.74 | 132.0 | 27.01 | 92.01 | 15.43 | 37.68 | 9.905 | 17.52 | 7.862 | 13.64 |
| | OSTQuant | GPTQ | 71.49 | 154.6 | 51.60 | 145.6 | 21.73 | 60.39 | 23.55 | 47.79 | 10.73 | 22.21 |
| | | BOA | 44.90 | 107.7 | 29.90 | 74.04 | 15.16 | 37.49 | 10.07 | 18.22 | 7.894 | 13.96 |
| | | TURBOBOA | 36.43 | 87.93 | 22.68 | 63.75 | 13.98 | 35.61 | 9.040 | 15.77 | 7.316 | 12.78 |
| W2A4KV4 | OmniQuant† | RTN | - | - | - | - | - | - | 1.0e5 | 1.9e5 | 3.8e3 | 5.4e3 |
| | DuQuant | RTN | 8.5e3 | 1.5e4 | 1.7e3 | 2.5e3 | 4.1e4 | 2.5e5 | 465.9 | 753.3 | 16.35 | 24.83 |
| | SpinQuant | GPTQ | 143.8 | 330.0 | 65.09 | 194.6 | 29.57 | 83.07 | 24.29 | 49.45 | 15.54 | 38.13 |
| | | BOA | 77.05 | 167.0 | 37.12 | 120.0 | 18.23 | 48.52 | 11.80 | 20.97 | 8.974 | 15.96 |
| | | TURBOBOA | 63.07 | 142.1 | 28.18 | 92.58 | 16.43 | 41.71 | 10.43 | 18.95 | 8.195 | 14.51 |
| | OSTQuant | GPTQ | 80.61 | 206.1 | 60.37 | 214.6 | 23.87 | 68.52 | 21.53 | 42.89 | 11.32 | 22.47 |
| | | BOA | 57.27 | 141.3 | 31.74 | 84.68 | 16.05 | 39.93 | 10.19 | 18.37 | 8.073 | 14.51 |
| | | TURBOBOA | 46.10 | 111.7 | 24.53 | 72.72 | 14.51 | 38.12 | 9.142 | 16.59 | 7.508 | 13.25 |

(b) Zero-shot Accuracy (↑)

| Precision | Transform | Quantizer | Llama3.2-1B | Llama3.2-3B | Llama3-8B | Llama2-7B | Llama2-13B |
|---|---|---|---|---|---|---|---|
| FP16 | Baseline | | 56.82 | 63.01 | 70.34 | 67.28 | 69.83 |
| W2A4KV16 | OmniQuant† | RTN | - | - | - | 30.63 | 30.19 |
| | DuQuant | RTN | 30.58 | 30.47 | 30.77 | 30.45 | 41.72 |
| | SpinQuant | GPTQ | 34.03 | 33.59 | 39.29 | 36.83 | 42.91 |
| | | BOA | 36.56 | 39.56 | 44.53 | 48.25 | 54.35 |
| | | TURBOBOA | 38.28 | 42.52 | 49.22 | 50.54 | 56.84 |
| | OSTQuant | GPTQ | 35.28 | 35.42 | 40.92 | 38.59 | 45.08 |
| | | BOA | 37.87 | 42.71 | 47.79 | 50.16 | 55.40 |
| | | TURBOBOA | 39.47 | 45.80 | 50.49 | 52.14 | 58.77 |
| W2A4KV4 | OmniQuant† | RTN | - | - | - | 30.29 | 29.78 |
| | DuQuant | RTN | 31.00 | 30.63 | 30.16 | 30.61 | 39.55 |
| | SpinQuant | GPTQ | 33.59 | 33.31 | 37.26 | 37.54 | 40.02 |
| | | BOA | 36.13 | 39.53 | 45.02 | 47.14 | 52.50 |
| | | TURBOBOA | 37.28 | 42.44 | 47.75 | 49.89 | 55.86 |
| | OSTQuant | GPTQ | 33.90 | 35.32 | 41.70 | 36.82 | 46.54 |
| | | BOA | 36.82 | 41.87 | 46.04 | 49.22 | 55.78 |
| | | TURBOBOA | 39.35 | 44.08 | 49.78 | 51.44 | 58.23 |

† The official code does not support models exploiting grouped query attention.

## 4.3 COMPARISON WITH PRIOR ARTS

We now compare the performance of TURBOBOA against existing LLM quantization methods. Our comparison includes BOA, which serves as the primary baseline, and transformation-based approaches that improve performance by suppressing outliers via scaling and/or rotation (*e.g.*, OmniQuant (Shao et al., 2023), DuQuant (Lin et al., 2024), QuaRot (Ashkboos et al., 2024), SpinQuant (Liu et al., 2024), and OSTQuant (Hu et al., 2025)); see Appendix B for the details of each method.

**Weight-only Quantization**   We first evaluate the performance of weight-only quantization. Following BOA (Kim et al., 2025), we integrate TURBOBOA with QuaRot, which requires no training and incurs no additional inference costs. The complementarity with other transformation-based approaches (*e.g.*, SpinQuant and OSTQuant) will be investigated in the weight-activation quantization setting (see Table 5). For results without any transformation, please refer to Appendix F.1, where we demonstrate the intrinsic effectiveness of the proposed error correction and grid selection. Notably, in Appendix F.2, we provide a direct comparison with GPTAQ (Li et al., 2025) to highlight the importance of incorporating dependencies between out-channels in low-bit regimes.

Table 4 summarizes the results under INT2 and INT3 quantization. Overall, BOA and the proposed TURBOBOA outperform other methods because they explicitly account for cross-layer dependencies within the attention module during weight quantization. In contrast, OmniQuant, SpinQuant-RTN,

and SpinQuant-GPTQ consider cross-layer dependencies only when learning transformation matrices and rely on naïve nearest rounding or GPTQ with layer-wise objectives, thereby failing to capture such dependencies during weight quantization. As shown, TURBOBOA consistently achieves the best results. For example, on 2-bit quantization of Llama3.2-1B, TURBOBOA improves Wiki2 PPL from 40.86 (BOA) to 33.33. The benefits extend to zero-shot evaluation as well, where TURBOBOA achieves at least 2%p accuracy gain over other methods across all model scales. Notably, under 3-bit quantization, TURBOBOA nearly preserves the FP performance. For instance, on Llama2-13B, TURBOBOA achieves 69.07%, which is very close to the FP baseline of 69.83%.

**Weight-Activation Quantization**   We next evaluate the performance of weight-activation quantization. Following prior works (Ashkboos et al., 2024; Liu et al., 2024; Kim et al., 2025), we quantize input activations to all linear layers and KV caches using the Min-Max quantizer, where quantization parameters are dynamically computed for each token. For outlier suppression, we integrate GPTQ, BOA, and TURBOBOA with either SpinQuant or OSTQuant. Unlike QuaRot, which relies on a fixed Hadamard matrix, SpinQuant and OSTQuant optimize rotation matrices by explicitly incorporating activation quantization effects during training (Liu et al., 2024; Hu et al., 2025).

Table 5 summarizes the results under W2A4KV4 and W2A4KV16 settings. Across both configurations and all model scales, TURBOBOA consistently outperforms BOA and other baselines. For example, with SpinQuant applied under W2A4KV4 on Llama3.2-1B, TURBOBOA reduces Wiki2 PPL from 77.05 (BOA) to 63.07. When combined with OSTQuant under W2A4KV16 on Llama3.2-3B, TURBOBOA lowers C4 PPL from 74.04 (BOA) to 63.75, while GPTQ and DuQuant exhibit substantially higher PPLs. Consistent gains are also observed for larger models such as Llama3-8B and Llama2-13B, confirming the scalability of the proposed approach. Beyond PPL, TURBOBOA delivers clear improvements in zero-shot accuracy. On Llama3-8B under W2A4KV16, TURBOBOA with SpinQuant achieves 49.22%, surpassing BOA by 5%p. On Llama2-13B under W2A4KV4, TURBOBOA with SpinQuant attains 55.86%, yielding an absolute gain of more than 3%p over BOA and over 15%p compared to GPTQ. These results demonstrate that TURBOBOA not only accelerates quantization but also achieves state-of-the-art performance in weight-activation quantization.

## 5   CONCLUSION

In this work, we proposed TURBOBOA, a backpropagation-free PTQ algorithm that addresses the key efficiency and accuracy bottlenecks of the conventional BOA. By quantizing multiple out-channels simultaneously, TURBOBOA significantly reduces sequential operations, accelerating the quantization process by more than three-fold. Furthermore, by extending error compensation to incorporate errors of previously quantized layers and adaptively determining quantization grids with a further CD-based refinement, TURBOBOA effectively mitigates error accumulation and misalignment, which could be critical in the low-bit regime. Our experimental results demonstrate that TURBOBOA delivers substantial speedup over BOA while achieving superior accuracy, and when combined with transformation-based outlier suppression methods, it establishes new state-of-the-art results in both weight-only and weight-activation quantization. We believe TURBOBOA paves the way for broader deployment of LLMs on resource-constrained hardware, offering a practical balance between computational efficiency and model fidelity.

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

## A  USE OF LLMS

In our work, LLMs were used solely to assist with paper writing, specifically for improving grammar, polishing phrasing, and enhancing readability. We have not used LLMs for developing research ideas, designing the methodology, conducting experiments, analyzing results, or drawing conclusions.

## B  TRANSFORMATION-BASED PTQ METHODS

As noted, transformation-based methods aim to suppress outliers within weights or activations by applying a certain type of transformation, such as scaling, rotation, and permutation. Transformation-based methods have often been used to improve the quantization robustness of models before conducting quantization (Ashkboos et al., 2024; Liu et al., 2024; Kim et al., 2025). If we denote a transformation matrix by $\mathbf{T}$, then the transformation in one layer can be expressed as

$$\mathbf{W}\mathbf{X} = (\mathbf{W}\mathbf{T})(\mathbf{T}^{-1}\mathbf{X}). \tag{10}$$

Under this formulation, the main goal of transformation-based methods is to construct a "good" $\mathbf{T}$ that makes $\mathbf{W}\mathbf{T}$ and $\mathbf{T}^{-1}\mathbf{X}$ easier to be quantized.

Over the years, various transformation-methods have been proposed. Each algorithm adopts a different strategy for constructing $\mathbf{T}$ to better suppress outliers and improve quantization robustness further. For example, some methods adopt lightweight deterministic transformations, while others learn $\mathbf{T}$ through optimization guided by calibration data. Below, we briefly summarize the contributions of each transformation-based method used in our comparison.

**OmniQuant** adopts a diagonal transformation matrix (*i.e.*, $\mathbf{T} = \mathrm{diag}(\mathbf{c})$) to mitigate activation outliers that persist in several channels across all tokens (Shao et al., 2023). The scaling factor $\mathbf{c}$ is jointly optimized with the quantization parameters (scale and zero-point) of each layer via gradient-based training. The learned scaling factor can be seamlessly merged into existing components (*e.g.*, normalization layers), thereby incurring no additional inference overhead. When measuring performance, we activated both learnable equivalent transformation (LET) and learnable weight clipping (LWC) options.

**QuaRot/SpinQuant** adopt orthogonal (rotation) matrices $\mathbf{R}$ (*i.e.*, $\mathbf{R}\mathbf{R}^T = \mathbf{R}^T\mathbf{R} = \mathbf{I}$), redistributing extremely large activation outliers that are present in few tokens (Ashkboos et al., 2024; Liu et al., 2024). While QuaRot employs Hadamard matrices as the orthogonal transformation and thus requires no training, SpinQuant learns $\mathbf{R}$ guided by calibration data. By applying the same orthogonal matrix across different Transformer layers, both methods can integrate $\mathbf{R}$ seamlessly into existing components, thereby incurring no inference overhead.

**DuQuant** integrates scaling $\mathrm{diag}(\mathbf{c})$, rotations $\mathbf{R}_1, \mathbf{R}_2$, and permutation $\mathbf{P}$ into a single transformation (*i.e.*, $\mathbf{T} = \mathrm{diag}(\mathbf{c})\mathbf{R}_1\mathbf{P}\mathbf{R}_2$) (Lin et al., 2024). It provides an efficient backpropagation-free algorithm to compute the transformation parameters $\mathbf{c}, \mathbf{R}_1, \mathbf{R}_2, \mathbf{P}$ for each layer. Unlike QuaRot and SpinQuant, DuQuant learns distinct parameters for different Transformation blocks. While this design incurs additional inference costs, the authors demonstrate through empirical timing measurements that the overhead remains manageable. When measuring performance, we activated the LWC option, which leads to the better result (Lin et al., 2024).

**OSTQuant** combines scaling and rotation for transformation (*i.e.*, $\mathbf{T} = \mathrm{diag}(\mathbf{c})\mathbf{R}$) (Hu et al., 2025). In addition, it introduces a new metric, termed quantization space utilization rate (QSUR), to evaluate the quantizability of transformed data and provides a theoretical justification that the joint use of scaling and rotation improves QSUR. The scaling factors and rotation matrices are learned through gradient-based training and then fused into the original inference graph.

## C  UPDATE RULE FOR ERROR COMPENSATION OF MULTIPLE OUT-CHANNELS

We solve the constrained optimization problem in (4) by exploiting its Lagrangian:

$$L(\Delta\mathbf{W}, \boldsymbol{\lambda}_0, \dots \boldsymbol{\lambda}_{N-1}) = \|\mathbf{G}\Delta\mathbf{W}\mathbf{X}\|_F^2 + \sum_{i=0}^{N-1}(\mathbf{e}_i^T\Delta\mathbf{W} - (\mathbf{Q}_{i,:} - \mathbf{W}_{i,:}))\boldsymbol{\lambda}_i$$

$$= \mathrm{tr}(\mathbf{H}_{out}\Delta\mathbf{W}\mathbf{H}_{in}\Delta\mathbf{W}^T) + \sum_{i=0}^{N-1}(\mathbf{e}_i^T\Delta\mathbf{W} + \mathbf{W}_{i,:} - \mathbf{Q}_{i,:})\boldsymbol{\lambda}_i,$$

where $\boldsymbol{\lambda}_0, \dots, \boldsymbol{\lambda}_{N-1} \in \mathbb{R}^{d_{in}\times 1}$ are Lagrange multipliers. Specifically, the update rule can be obtained by taking derivatives of the Lagrangian $L(\Delta\mathbf{W}, \boldsymbol{\lambda}_0, \dots \boldsymbol{\lambda}_{N-1})$ and then setting these derivatives to zero:

$$\frac{\partial L}{\partial \Delta\mathbf{W}} = 2\mathbf{H}_{out}\Delta\mathbf{W}\mathbf{H}_{in} + \sum_{i=0}^{N-1}\mathbf{e}_i\boldsymbol{\lambda}_j^T = \mathbf{0}_{d_{out}\times d_{in}}, \tag{11a}$$

$$\begin{bmatrix} (\partial L/\partial\boldsymbol{\lambda}_0)^T \\ \vdots \\ (\partial L/\partial\boldsymbol{\lambda}_{N-1})^T \end{bmatrix} = \begin{bmatrix} \mathbf{e}_0^T\Delta\mathbf{W} + \mathbf{W}_{0,:} - \mathbf{Q}_{0,:} \\ \vdots \\ \mathbf{e}_{N-1}^T\Delta\mathbf{W} + \mathbf{W}_{N-1,:} - \mathbf{Q}_{N-1,:} \end{bmatrix} = [\Delta\mathbf{W}]_{B,:} + (\mathbf{W}_{B,:} - \mathbf{Q}_{B,:}) = \mathbf{0}_{N\times d_{in}} \tag{11b}$$

As a result, the solution is attained when

$$\Delta\mathbf{W} = -\frac{1}{2}[\mathbf{H}_{out}^{-1}]_{:,B}\begin{bmatrix} \boldsymbol{\lambda}_0^T \\ \vdots \\ \boldsymbol{\lambda}_{N-1}^T \end{bmatrix}\mathbf{H}_{in}^{-1}, \tag{12a}$$

$$[\Delta\mathbf{W}]_{B,:} = -(\mathbf{W}_{B,:} - \mathbf{Q}_{B,:}), \tag{12b}$$

combining which yields

$$\begin{bmatrix} \boldsymbol{\lambda}_0^T \\ \vdots \\ \boldsymbol{\lambda}_{N-1}^T \end{bmatrix} = 2[\mathbf{H}_{out}^{-1}]_{B,B}^{-1}(\mathbf{W}_{B,:} - \mathbf{Q}_{B,:})\mathbf{H}_{in}. \tag{13}$$

Finally, by combining (12a) and (13), we obtain the desired update rule in (5):

$$[\Delta\mathbf{W}]_{N:,:} = -[\mathbf{H}_{out}^{-1}]_{N:,B}[\mathbf{H}_{out}^{-1}]_{B,B}^{-1}(\mathbf{W}_{B,:} - \mathbf{Q}_{B,:})$$

$$\overset{(a)}{=} -[\mathbf{U}_{out}^T\mathbf{U}_{out}]_{N:,B}[\mathbf{U}_{out}^T\mathbf{U}_{out}]_{B,B}^{-1}(\mathbf{W}_{B,:} - \mathbf{Q}_{B,:})$$

$$\overset{(b)}{=} -[\mathbf{U}_{out}^T]_{N:,B}[\mathbf{U}_{out}]_{B,B}\left([\mathbf{U}_{out}^T]_{B,B}[\mathbf{U}_{out}]_{B,B}\right)^{-1}(\mathbf{W}_{B,:} - \mathbf{Q}_{B,:})$$

$$= -[\mathbf{U}_{out}^T]_{N:,B}[\mathbf{U}_{out}^T]_{B,B}^{-1}(\mathbf{W}_{B,:} - \mathbf{Q}_{B,:}), \tag{14}$$

where (a) is because $\mathbf{U}_{out} = \mathrm{Chol}(\mathbf{H}_{out}^{-1})^T$ and (b) is because $\mathbf{U}_{out}$ is upper triangular.

## D UPDATE RULE INCORPORATING QUANTIZATION ERRORS OF EARLIER TRANSFORMER BLOCKS

We note that the objective function (7) can be expressed as

$$
\begin{aligned}
\|\mathbf{G}\Delta\mathbf{W}\mathbf{X} + \mathbf{G}_{:,B}\mathbf{W}_{B,:}\Delta\mathbf{X}\|_F^2 &= \|\mathbf{G}\Delta\mathbf{W}\mathbf{X}\|_F^2 + 2\mathrm{tr}(\mathbf{G}_{:,B}\mathbf{W}_{B,:}\Delta\mathbf{X}\mathbf{X}^T\Delta\mathbf{W}^T\mathbf{G}^T) + c \\
&= \|\mathbf{G}\Delta\mathbf{W}\mathbf{X}\|_F^2 + 2\mathrm{tr}(\mathbf{G}^T\mathbf{G}_{:,B}\mathbf{W}_{B,:}\Delta\mathbf{X}\mathbf{X}^T\Delta\mathbf{W}^T) + c \\
&= \|\mathbf{G}\Delta\mathbf{W}\mathbf{X}\|_F^2 + 2\mathrm{tr}([\mathbf{H}_{out}]_{:,B}\mathbf{W}_{B,:}\mathbf{R}\Delta\mathbf{W}^T) + c,
\end{aligned}
$$

where $c = \|\mathbf{G}_{:,B}\mathbf{W}_{B,:}\Delta\mathbf{X}\|_F^2$ is constant with respect to $\Delta\mathbf{W}$ and the last equality holds because $\mathbf{H}_{out} = \mathbf{G}^T\mathbf{G}$ and $\mathbf{R} = \Delta\mathbf{X}\mathbf{X}^T$. Thus, the optimization problem in (7) is equivalent to

$$
\begin{aligned}
\min_{\Delta\mathbf{W}} \quad & \|\mathbf{G}\Delta\mathbf{W}\mathbf{X}\|_F^2 + 2\mathrm{tr}([\mathbf{H}_{out}]_{:,B}\mathbf{W}_{B,:}\mathbf{R}\Delta\mathbf{W}^T) \\
\mathrm{s.t.} \quad & \mathbf{e}_i^T\Delta\mathbf{W} = \mathbf{Q}_{i,:} - \mathbf{W}_{i,:} \; (i \in B).
\end{aligned} \tag{15}
$$

Compared to the optimization problem in (4) for the first layer, it involves an additional term in the objective whose derivative is

$$
\frac{\partial}{\partial\Delta\mathbf{W}}\left(2\mathrm{tr}([\mathbf{H}_{out}]_{:,B}\mathbf{W}_{B,:}\mathbf{R}\Delta\mathbf{W}^T)\right) = 2[\mathbf{H}_{out}]_{:,B}\mathbf{W}_{B,:}\mathbf{R}. \tag{16}
$$

Using this together with (11a), the solution is attained when

$$
\frac{\partial L}{\partial\Delta\mathbf{W}} = 2\mathbf{H}_{out}\Delta\mathbf{W}\mathbf{H}_{in} + \sum_{i=0}^{N-1}\mathbf{e}_i\boldsymbol{\lambda}_j^T + 2[\mathbf{H}_{out}]_{:,B}\mathbf{W}_{B,:}\mathbf{R} = \mathbf{0}_{d_{out}\times d_{in}},
$$

which is equivalent to

$$
\Delta\mathbf{W} = -\frac{1}{2}[\mathbf{H}_{out}^{-1}]_{:,B}\begin{bmatrix}\boldsymbol{\lambda}_0^T \\ \vdots \\ \boldsymbol{\lambda}_{N-1}^T\end{bmatrix}\mathbf{H}_{in}^{-1} - \mathbf{I}_{:,B}\mathbf{W}_{B,:}\mathbf{R}\mathbf{H}_{in}^{-1}. \tag{17}
$$

Combining this with (12b) yields

$$
[\Delta\mathbf{W}]_{B,:} = -\frac{1}{2}[\mathbf{H}_{out}^{-1}]_{B,B}\begin{bmatrix}\boldsymbol{\lambda}_0^T \\ \vdots \\ \boldsymbol{\lambda}_{N-1}^T\end{bmatrix}\mathbf{H}_{in}^{-1} - \mathbf{W}_{B,:}\mathbf{R}\mathbf{H}_{in}^{-1} = -(\mathbf{W}_{B,:} - \mathbf{Q}_{B,:}),
$$

which leads to

$$
\begin{bmatrix}\boldsymbol{\lambda}_0^T \\ \vdots \\ \boldsymbol{\lambda}_{N-1}^T\end{bmatrix} = 2[\mathbf{H}_{out}^{-1}]_{B,B}^{-1}(\mathbf{W}_{B,:} - \mathbf{Q}_{B,:})\mathbf{H}_{in} - 2[\mathbf{H}_{out}^{-1}]_{B,B}^{-1}\mathbf{W}_{B,:}\mathbf{R}. \tag{18}
$$

Finally, by combining (17) and (18), we obtain the desired update rule in (8):

$$
\begin{aligned}
[\Delta\mathbf{W}]_{N,:} &= -[\mathbf{H}_{out}^{-1}]_{N,B}[\mathbf{H}_{out}^{-1}]_{B,B}^{-1}(\mathbf{W}_{B,:} - \mathbf{Q}_{B,:}) - (\mathbf{I}_{N,:B} - [\mathbf{H}_{out}^{-1}]_{N,:B}[\mathbf{H}_{out}^{-1}]_{B,B}^{-1})\mathbf{W}_{B,:}\mathbf{R}\mathbf{H}_{in}^{-1} \\
&= -[\mathbf{H}_{out}^{-1}]_{N,B}[\mathbf{H}_{out}^{-1}]_{B,B}^{-1}(\mathbf{W}_{B,:} - \mathbf{Q}_{B,:}) + [\mathbf{H}_{out}^{-1}]_{N,:B}[\mathbf{H}_{out}^{-1}]_{B,B}^{-1}\mathbf{W}_{B,:}\mathbf{R}\mathbf{H}_{in}^{-1} \\
&= -[\mathbf{U}_{out}^T]_{N,:B}[\mathbf{U}_{out}^T]_{B,B}^{-1}(\mathbf{W}_{B,:} - \mathbf{Q}_{B,:}) + [\mathbf{U}_{out}^T]_{N,:B}[\mathbf{U}_{out}^T]_{B,B}^{-1}\mathbf{W}_{B,:}\mathbf{R}\mathbf{H}_{in}^{-1},
\end{aligned}
$$

where the last equality holds because $\mathbf{U}_{out} = \mathrm{Chol}(\mathbf{H}_{out}^{-1})^T$ (see (a) and (b) in Appendix C).

# E    ATTENTION-AWARE SCALE REFINEMENT VIA CD

Let $\mathbf{G} = [\mathbf{g}_0 \cdots \mathbf{g}_{d_{out}-1}]$ and $\mathbf{W}_{int} = [\mathbf{w}_{int,0} \cdots \mathbf{w}_{int,d_{out}-1}]^T$, then the attention reconstruction error in (9) is expressed as

$$
\begin{aligned}
\mathcal{L}(\mathbf{s}) &= \|\mathbf{G}\,\mathrm{diag}(\mathbf{s})\mathbf{W}_{int}\mathbf{X}\|_F^2 - 2\langle \mathbf{G}\,\mathrm{diag}(\mathbf{s})\mathbf{W}_{int}\mathbf{X}, \mathbf{G}\mathbf{W}\widetilde{\mathbf{X}}\rangle_F + c \\
&= \left\|\sum_{j=0}^{d_{out}-1} s_j \mathbf{g}_j \mathbf{w}_{int,j}^T \mathbf{X}\right\|_F^2 - 2\sum_{j=0}^{d_{out}-1}\langle s_j \mathbf{g}_j \mathbf{w}_{int,j}^T \mathbf{X}, \mathbf{G}\mathbf{W}\widetilde{\mathbf{X}}\rangle_F + c \\
&= \sum_{j,k=0}^{d_{out}-1} \mathrm{tr}(\mathbf{g}_j \mathbf{w}_{int,j}^T \mathbf{X}\mathbf{X}^T \mathbf{w}_{int,k}\mathbf{g}_k^T)s_j s_k - 2\sum_{j=0}^{d_{out}-1}\mathrm{tr}(\mathbf{g}_j \mathbf{w}_{int,j}^T \mathbf{X}\widetilde{\mathbf{X}}^T \mathbf{W}^T \mathbf{G}^T)s_j + c,
\end{aligned}
$$

where $\langle\cdot,\cdot\rangle_F$ denotes the Frobenius inner product (i.e., $\langle\mathbf{A}, \mathbf{B}\rangle_F = \mathrm{tr}(\mathbf{A}\mathbf{B}^T)$) and $c$ is constant with respect to scales $\mathbf{s}$. Using this together with $\mathbf{H}_{in} = \mathbf{X}\mathbf{X}^T$ and $\mathbf{X}\widetilde{\mathbf{X}}^T = \mathbf{X}(\mathbf{X} - \Delta\mathbf{X})^T = (\mathbf{H}_{in} - \mathbf{R})^T$, we have

$$
\begin{aligned}
\mathcal{L}(\mathbf{s}) &= \sum_{j,k=0}^{d_{out}-1}\mathrm{tr}(\mathbf{g}_j \mathbf{w}_{int,j}^T \mathbf{H}_{in}\mathbf{w}_{int,k}\mathbf{g}_k^T)s_j s_k - 2\sum_{j=0}^{d_{out}-1}\mathrm{tr}(\mathbf{g}_j \mathbf{w}_{int,j}^T (\mathbf{H}_{in} - \mathbf{R})^T \mathbf{W}^T \mathbf{G}^T)s_j + c \\
&= \sum_{j,k=0}^{d_{out}-1}(\mathbf{w}_{int,j}^T \mathbf{H}_{in}\mathbf{w}_{int,k}\mathbf{g}_k^T \mathbf{g}_j)s_j s_k - 2\sum_{j=0}^{d_{out}-1}(\mathbf{w}_{int,j}^T (\mathbf{H}_{in} - \mathbf{R})^T \mathbf{W}^T \mathbf{G}^T \mathbf{g}_j)s_j + c \\
&= \sum_{j,k=0}^{d_{out}-1}[\mathbf{W}_{int}\mathbf{H}_{in}\mathbf{W}_{int}^T]_{j,k}[\mathbf{H}_{out}]_{k,j}\cdot s_j s_k - 2\sum_{j=0}^{d_{out}-1}[\mathbf{W}_{int}(\mathbf{H}_{in} - \mathbf{R})^T \mathbf{W}^T \mathbf{H}_{out}]_{j,j}\cdot s_j + c,
\end{aligned}
$$

where the last equality holds because $\mathbf{H}_{out} = \mathbf{G}^T\mathbf{G}$. To minimize $\mathcal{L}(\mathbf{s})$, we adopt the CD algorithm, i.e., we iteratively update one scale at a time while keeping the others fixed. Since the loss is quadratic in $s_j$, the update formula for $s_j$ can be obtained by setting $\partial\mathcal{L}/\partial s_j = 0$:

$$
\begin{aligned}
s_j^* &= \frac{[\mathbf{W}_{int}(\mathbf{H}_{in} - \mathbf{R})^T \mathbf{W}^T \mathbf{H}_{out}]_{j,j} - \sum_{k\neq j}[\mathbf{W}_{int}\mathbf{H}_{in}\mathbf{W}_{int}^T]_{j,k}[\mathbf{H}_{out}]_{k,j}s_k}{[\mathbf{W}_{int}\mathbf{H}_{in}\mathbf{W}_{int}^T]_{j,j}[\mathbf{H}_{out}]_{j,j}} \\
&= s_j + \frac{[\mathbf{W}_{int}(\mathbf{H}_{in} - \mathbf{R})^T \mathbf{W}^T \mathbf{H}_{out}]_{j,j} - \sum_k[\mathbf{W}_{int}\mathbf{H}_{in}\mathbf{W}_{int}^T]_{j,k}[\mathbf{H}_{out}]_{k,j}s_k}{[\mathbf{W}_{int}\mathbf{H}_{in}\mathbf{W}_{int}^T]_{j,j}[\mathbf{H}_{out}]_{j,j}} \\
&= s_j + \frac{[\mathbf{W}_{int}(\mathbf{H}_{in} - \mathbf{R})^T \mathbf{W}^T \mathbf{H}_{out}]_{j,j} - [\mathbf{W}_{int}\mathbf{H}_{in}\mathbf{Q}^T \mathbf{H}_{out}]_{j,j}}{[\mathbf{W}_{int}\mathbf{H}_{in}\mathbf{W}_{int}^T]_{j,j}[\mathbf{H}_{out}]_{j,j}} \\
&= s_j + \frac{[\mathbf{W}_{int}(\mathbf{H}_{in}(\mathbf{W} - \mathbf{Q})^T - \mathbf{R}^T \mathbf{W}^T)\mathbf{H}_{out}]_{j,j}}{[\mathbf{W}_{int}\mathbf{H}_{in}\mathbf{W}_{int}^T]_{j,j}[\mathbf{H}_{out}]_{j,j}},
\end{aligned}
$$

which completes the proof. In Algorithm 2, we summarize the pseudocode for the CD-based scale refinement.

---

**Algorithm 2** Coordinate Descent-based Scale Refinement

---

**Input**: FP weights $\mathbf{W}$, integer weights $\mathbf{W}_{int}$, initial scales $\mathbf{s}$, Hessians $\mathbf{H}_{out}$ and $\mathbf{H}_{in}$, and deviation correlation $\mathbf{R} = \Delta\mathbf{X}\mathbf{X}^T$
**Output**: refined scales $\mathbf{s}$
 1: Initialize quantized weights: $\mathbf{Q} \leftarrow \mathrm{diag}(\mathbf{s})\mathbf{W}_{int}$
 2: **for** $\ell = 0, \cdots, n_{iter} - 1$ **do**
 3:    **for** $j = 0, \cdots, d_{out} - 1$ **do**
 4:        Update scale for the $j$-th out-channel:

$$
s_j \leftarrow s_j + \frac{[\mathbf{W}_{int}(\mathbf{H}_{in}(\mathbf{W} - \mathbf{Q})^T - \mathbf{R}^T \mathbf{W}^T)\mathbf{H}_{out}]_{j,j}}{[\mathbf{W}_{int}\mathbf{H}_{in}\mathbf{W}_{int}^T]_{j,j}[\mathbf{H}_{out}]_{j,j}}
$$

 5:        Update quantized weights: $\mathbf{Q} \leftarrow \mathrm{diag}(\mathbf{s})\mathbf{W}_{int}$

---

# F  ADDITIONAL EXPERIMENTAL RESULTS

In this appendix, we present supplementary experimental results that were omitted from the main text due to page constraints. Specifically, we provide (i) weight-only quantization results without applying any transformations (*e.g.*, scaling or rotation), (ii) a direct comparison with GPTAQ, (iii) weight-activation quantization results under higher weight bit-widths, and (iv) an ablation study on the number of CD iterations.

## F.1  WEIGHT-ONLY QUANTIZATION PERFORMANCE WITHOUT TRANSFORMATION

Table 6 reports the weight-only quantization performance of GPTQ, BOA, and the proposed TUR-BOBOA without additional transformation. Across both 2-bit and 3-bit settings, TURBOBOA consistently outperforms both GPTQ and BOA. For instance, on Llama3.2-1B (INT2), TURBOBOA significantly reduces Wiki2 PPL from 538.9 (GPTQ) and 312.2 (BOA) to 111.3, while simultaneously improving zero-shot accuracy by 2.5%p. Even under the INT3 setting, TURBOBOA achieves clear improvements over BOA, demonstrating that the proposed enhancements remain highly effective even in the absence of transformation-based outlier suppression.

Table 6: Weight-only quantization performance on Llama2 and Llama3 models

(a) Wiki2 PPL (↓)

| Precision | Method | Llama3.2-1B | Llama3.2-3B | Llama3-8B | Llama2-7B | Llama2-13B |
|---|---|---|---|---|---|---|
| FP16 | Baseline | 13.16 | 11.05 | 6.139 | 5.473 | 4.885 |
| INT3 | RTN | 1.9e3 | 882.6 | 129.1 | 342.4 | 227.2 |
| | GPTQ | 112.0 | 46.14 | 8.226 | 6.719 | 9.790 |
| | BOA | 26.43 | 13.64 | 7.782 | 6.007 | 5.833 |
| | **TURBOBOA** | **19.73** | **13.12** | **7.523** | **5.958** | **5.288** |
| INT2 | RTN | 6.3e4 | 2.0e4 | 6.6e4 | 7.7e3 | 5.7e3 |
| | GPTQ | 538.9 | 98.19 | 24.54 | 30.85 | 35.08 |
| | BOA | 312.2 | 54.64 | 21.70 | 12.76 | 18.33 |
| | **TURBOBOA** | **111.3** | **33.42** | **17.83** | **9.781** | **13.09** |

(b) C4 PPL (↓)

| Precision | Method | Llama3.2-1B | Llama3.2-3B | Llama3-8B | Llama2-7B | Llama2-13B |
|---|---|---|---|---|---|---|
| FP16 | Baseline | 21.31 | 16.49 | 9.444 | 7.266 | 6.730 |
| INT3 | RTN | 1.6e3 | 736.1 | 119.8 | 2.7e3 | 245.0 |
| | GPTQ | 201.2 | 150.8 | 20.05 | 92.15 | 20.17 |
| | BOA | 37.98 | 24.05 | 14.10 | 8.686 | 7.634 |
| | **TURBOBOA** | **36.43** | **23.79** | **13.59** | **8.554** | **7.587** |
| INT2 | RTN | 4.6e4 | 1.1e4 | 8.2e4 | 8.2e3 | 4.8e3 |
| | GPTQ | 1.2e3 | 413.8 | 214.3 | 321.1 | 97.52 |
| | BOA | 571.9 | 214.0 | 92.69 | 26.42 | 28.36 |
| | **TURBOBOA** | **313.8** | **166.6** | **81.24** | **17.66** | **19.89** |

(c) Zero-shot Accuracy (↑)

| Precision | Method | Llama3.2-1B | Llama3.2-3B | Llama3-8B | Llama2-7B | Llama2-13B |
|---|---|---|---|---|---|---|
| FP16 | Baseline | 56.82 | 63.01 | 70.34 | 67.28 | 69.83 |
| INT3 | RTN | 33.19 | 33.37 | 36.01 | 33.18 | 32.92 |
| | GPTQ | 37.44 | 39.19 | 61.72 | 58.38 | 54.84 |
| | BOA | 47.05 | 59.38 | 65.37 | 63.70 | 63.35 |
| | **TURBOBOA** | **47.46** | **59.67** | **67.07** | **64.17** | **67.24** |
| INT2 | RTN | 31.08 | 30.99 | 32.79 | 30.19 | 30.12 |
| | GPTQ | 30.48 | 34.39 | 36.01 | 42.50 | 39.08 |
| | BOA | 31.33 | 38.53 | 42.16 | 45.81 | 45.06 |
| | **TURBOBOA** | **33.91** | **42.03** | **44.87** | **51.41** | **47.35** |

## F.2  COMPARISON WITH GPTAQ

To further validate the importance of incorporating inter-channel dependencies, we provide a direct comparison between GPTAQ (Li et al., 2025) and TURBOBOA. While both algorithms aim to compensate for quantization errors from preceding layers, they differ fundamentally in their treatment of the out-channel-wise Hessian $\mathbf{H}_{out}$. Specifically, while GPTAQ assumes $\mathbf{H}_{out} = \mathbf{I}$, thereby ignoring the correlations between out-channels, the proposed TURBOBOA explicitly incorporates the attention-aware Hessian $\mathbf{H}_{out}$ (see Table 1) to capture these dependencies.

Table 7 reports the performance on Llama3 models under the 2-bit weight-only quantization setting without additional transformations. Across all model scales, TURBOBOA consistently outperforms GPTAQ in both PPL and zero-shot accuracy. Notably, TURBOBOA achieves a 7.7%p accuracy gain on Llama-3.2-3B and a 10.5%p improvement on Llama-3-8B compared to GPTAQ. These results highlight that accounting for inter-channel dependencies is crucial for mitigating accuracy degradation in aggressive low-bit regimes.

Table 7: Evaluation on Llama3 models (INT2 quantization)

| Method | Llama3.2-1b | | Llama3.2-3b | | Llama3-8b | |
|---|---|---|---|---|---|---|
| | Wiki2 ($\downarrow$) | 0-shot ($\uparrow$) | Wiki2 ($\downarrow$) | 0-shot ($\uparrow$) | Wiki2 ($\downarrow$) | 0-shot ($\uparrow$) |
| GPTAQ | 200.5 | 31.73 | 47.90 | 34.31 | 19.29 | 34.36 |
| **TURBOBOA** | **111.3** | **33.91** | **33.42** | **42.03** | **17.83** | **44.87** |

### F.3 Weight-Activation Quantization Performance under Higher Bit-widths

We further report weight-activation quantization results under higher weight bit-widths in Table 8. In this table, results for OmniQuant are excluded because its official implementation does not support models utilizing grouped query attention. As expected, the performance gap among different algorithms narrows in this regime, as 4-bit quantization preserves most of the original FP accuracy. Nevertheless, TurboBoA consistently provides robust improvements in almost all cases, confirming the effectiveness of our method even when quantization is less challenging.

Table 8: Weight-activation quantization performance on transformed Llama3 models

(a) PPL (↓)

| Precision | Transform | Quantizer | Llama3.2-1B | | Llama3.2-3B | | Llama3-8B | |
|---|---|---|---|---|---|---|---|---|
| | | | Wiki2 | C4 | Wiki2 | C4 | Wiki2 | C4 |
| FP16 | Baseline | | 13.16 | 21.31 | 11.05 | 16.49 | 6.139 | 9.444 |
| W4A4KV16 | DuQuant | RTN | 1.9e4 | 1.8e4 | 13.32 | 19.49 | 8.066 | 13.24 |
| | SpinQuant | GPTQ | 16.68 | 26.87 | 11.87 | 19.47 | 7.636 | 12.59 |
| | | BoA | 16.25 | 26.29 | 11.57 | **19.04** | 7.496 | 12.35 |
| | | **TurboBoA** | **16.09** | **26.12** | **11.55** | 19.11 | **7.474** | **12.32** |
| | OSTQuant | GPTQ | 16.02 | 25.26 | 11.88 | 18.60 | 7.349 | 12.04 |
| | | BoA | 15.60 | 24.81 | 11.74 | 18.45 | 7.224 | 11.82 |
| | | **TurboBoA** | **15.53** | **24.68** | **11.69** | **18.31** | **7.213** | **11.78** |
| W4A4KV4 | DuQuant | RTN | 1.7e4 | 1.4e4 | 13.84 | 20.52 | 8.402 | 13.59 |
| | SpinQuant | GPTQ | 18.31 | 29.46 | 12.24 | 20.21 | 7.869 | 12.99 |
| | | BoA | 17.83 | 28.65 | 11.98 | 19.84 | 7.705 | 12.75 |
| | | **TurboBoA** | **17.77** | **28.56** | **11.88** | **19.73** | **7.680** | **12.70** |
| | OSTQuant | GPTQ | 17.29 | 28.19 | 12.64 | 20.00 | 7.540 | 12.42 |
| | | BoA | 16.89 | 27.30 | 12.43 | 19.58 | 7.428 | 12.22 |
| | | **TurboBoA** | **16.86** | **27.10** | **12.39** | **19.45** | **7.416** | **12.20** |

(b) Zero-shot Accuracy (↑)

| Precision | Transform | Quantizer | Llama3.2-1B | Llama3.2-3B | Llama3-8B |
|---|---|---|---|---|---|
| FP16 | Baseline | | 56.82 | 63.01 | 70.34 |
| W4A4KV16 | DuQuant | RTN | 30.33 | 57.93 | 63.15 |
| | SpinQuant | GPTQ | 50.89 | 58.71 | 64.79 |
| | | BoA | 51.76 | 59.17 | 65.31 |
| | | **TurboBoA** | **52.32** | **59.42** | **66.15** |
| | OSTQuant | GPTQ | 52.48 | 60.16 | 66.66 |
| | | BoA | 53.24 | 60.94 | 67.43 |
| | | **TurboBoA** | **53.67** | **61.65** | **67.88** |
| W4A4KV4 | DuQuant | RTN | 30.71 | 56.53 | 62.76 |
| | SpinQuant | GPTQ | 48.86 | 57.54 | 64.05 |
| | | BoA | 50.41 | **58.90** | 65.03 |
| | | **TurboBoA** | **50.73** | 58.77 | **65.64** |
| | OSTQuant | GPTQ | 50.44 | 59.34 | 65.25 |
| | | BoA | 50.94 | 59.66 | 66.47 |
| | | **TurboBoA** | **51.54** | **59.86** | **66.73** |

## F.4 Ablation on the number of CD iterations

In this subsection, we investigate the impact of the number $n_{iter}$ of CD iterations (see Algorithm 2) on quantization quality. We focus on the attention reconstruction loss $\|\mathbf{G}\Delta\mathbf{W}\mathbf{X}\|_F^2$ measured at the first Transformer block to avoid confounding effects from previous blocks. The results in Table 9 indicate that the first CD iteration accounts for nearly all the reduction in loss, with additional iterations yielding diminishing returns. Accordingly, the end-to-end PPL performance remains virtually unchanged between 1 and 2 iterations. To maintain optimal computational efficiency, we set the CD iteration count to 1 for all main experiments.

Table 9: Ablation on the number of CD iterations

| Model | $n_{iter}$ | Loss (Query) | Loss (Key) | Wiki2 ($\downarrow$) | C4 ($\downarrow$) |
|---|---|---|---|---|---|
| | 0 | 317.6 | 66.97 | 37.15 | 92.58 |
| Llama3.2-1B | 1 | 315.9 | 66.68 | 33.33 | 85.55 |
| | 2 | 315.8 | 66.67 | 32.28 | 88.03 |
| | 0 | 170.1 | 70.72 | 25.92 | 63.48 |
| Llama3.2-3B | 1 | 168.9 | 70.25 | 24.10 | 54.20 |
| | 2 | 168.7 | 70.16 | 24.07 | 54.53 |
| | 0 | 126.1 | 43.36 | 14.21 | 34.67 |
| Llama3-8B | 1 | 125.6 | 43.20 | 13.54 | 32.99 |
| | 2 | 125.5 | 43.17 | 13.46 | 33.23 |

## G PSEUDOCODE FOR GPTAQ

In this appendix, we provide the pseudocode of the conventional GPTAQ (Li et al., 2025), which is omitted in the main manuscript due to the page limitation.

---

**Algorithm 3** GPTAQ

---

**Input**: weights $\mathbf{W}$, Hessian information $\mathbf{U}_{in}$, deviation correlation $\mathbf{R} = \Delta\mathbf{X}\mathbf{X}^T$, and scale $\mathbf{s}$
1: Initialize quantized and integer weights: $\mathbf{Q}, \mathbf{W}_{int} \leftarrow \mathbf{0}_{d_{out} \times d_{in}}$
2: Compute $\mathbf{P} = \left(\mathbf{R}\mathbf{U}_{in}^T \odot \mathbf{M_U}\right)\mathbf{U}_{in}$ ($\mathbf{M_U}$: strictly upper triangular masking matrix with ones above the diagonal)
3: **for** $j = 0, \cdots, d_{in} - 1$ **do**
4:     Quantize the $j$-th in-channel:
$$[\mathbf{W}_{int}]_{:,j} \leftarrow \text{clamp}\left(\left\lfloor \text{diag}(\mathbf{s})^{-1}\mathbf{W}_{:,j}\right\rceil, 0, 2^b - 1\right)$$
$$\mathbf{Q}_{:,j} \leftarrow \text{diag}(\mathbf{s})[\mathbf{W}_{int}]_{:,j}$$
5:     Estimate quantization error: $\mathbf{E}_{:,j} \leftarrow (\mathbf{W}_{:,j} - \mathbf{Q}_{:,j})/[\mathbf{U}_{in}]_{j,j}$
6:     Update remaining in-channels:
$$\mathbf{W}_{:,j:} \leftarrow \mathbf{W}_{:,j:} - \frac{\mathbf{W}_{:,j} - \mathbf{Q}_{:,j}}{[\mathbf{U}_{in}]_{j,j}}[\mathbf{U}_{in}]_{j,j:} - \mathbf{W}_{:,j}\mathbf{P}_{j,j:}$$
**Output**: quantized weights $\mathbf{Q}$, integer weights $\mathbf{W}_{int}$

---

