# OpenReview forum: "TurboBoA: Faster and Exact Attention-aware Quantization without Backpropagation"
_ICLR.cc/2026/Conference — ICLR 2026 Poster_

### Official Review · Reviewer_MvrE · 2025-10-30

**Soundness:** 3
**Presentation:** 3
**Contribution:** 3
**Rating:** 6
**Confidence:** 4

**Summary:**

This paper addresses the challenge of efficiently performing post-training quantization for Transformers, especially attention layers, without backpropagation, while maintaining high accuracy at extremely low bit-widths. It proposes TURBOBOA, which jointly quantizes multiple output channels with closed-form error compensation, incorporates cross-layer error correction, and adaptively refines quantization grids. Experiments on LLaMA models show that TURBOBOA achieves 4–6× speedup over BOA while improving perplexity and zero-shot accuracy, particularly in 2-bit and 3-bit settings. The results demonstrate that TURBOBOA effectively balances efficiency and precision, making low-bit quantization more practical for large-scale Transformers.

**Strengths:**

- The paper clearly identifies the limitations of existing post-training quantization methods, particularly the trade-off between accuracy and efficiency in attention-aware quantization.

- TURBOBOA introduces joint channel quantization with closed-form error compensation, cross-layer error correction, and adaptive grid refinement, effectively addressing accuracy loss in low-bit quantization.

- The method is evaluated on multiple LLaMA models with various bit-widths and settings, showing significant speedup and improved perplexity and zero-shot accuracy.

- The paper successfully demonstrates that TURBOBOA achieves a practical trade-off between computational efficiency and quantization accuracy for large-scale Transformer models.

**Weaknesses:**

- The method involves multiple closed-form derivations and joint updates, which may increase implementation and debugging difficulty.

- Cross-layer error compensation requires extra intermediate computations, which could be non-negligible on very large models or resource-constrained hardware.

- The choice of the number of jointly quantized channels (N) can affect both speed and accuracy, and may require tuning for different models or layers.

- This paper emphasizes that backpropagation is not required, but in practice it still involves a considerable amount of computation. In contrast, EfficientQAT[1] achieves significant performance improvements with minimal cost. Could you elaborate again on the advantages of being backpropagation-free? Generally, we still aim to obtain models with higher accuracy benefits.

[1] Chen, Mengzhao, et al. "Efficientqat: Efficient quantization-aware training for large language models." arXiv preprint arXiv:2407.11062 (2024).

**Questions:**

Please refer to the weaknesses above.

---

> ### Author Response · Authors · 2025-11-21
>
> **1. Overhead caused by cross-layer error compensation**
> - As the reviewer pointed out, cross-layer error compensation (F2) introduces non-negligible runtime overhead. In fact, as shown in Table 3, incorporating F2 increases runtime from 24.59 to 37.83 minutes for Llama3-8B, while adding other feature (F3) introduces only a marginal overhead (from 24.59 to 25.99 minutes).
> - The main reason for the non-neligible overhead of F2 is that the activations $\widetilde{\mathbf{X}}$ for the original full-precision (FP) model must also be computed when F2 is applied, which requires a forward pass over the entire model. However, we emphasize that despite this overhead, TurboBoA achieves a 2.5–3.5× overall speedup over BoA, thanks to the reduced number of sequential operations (see Table 3). We will clarify this behavior in the revised manuscript.
> - Finally, we note that in scenarios where memory or time resources are limited, F2 can be omitted. In these cases, by applying only F1, TurboBoA achieves a performance close to BoA but with a significant reduction in processing time (see Table I). This trade-off provides an efficient option when F2 is not feasible due to hardware or time constraints. We will include this in the revised manuscript.
>
> **2. Expanding the analysis of $N$**
> - To address the reviewer's concern, we expanded our ablations with $N \in [4, 8, 16, 32, 64]$. We note that $N=128(=d_{h})$ corresponds to GPTQ.
> - From the results in Table I, we observe a significant speed improvement up to $N=16$, while further increasing $N$ to 32 or 64 yields only marginal additional gains. This is because the cost of the inverse operation (see $([\mathbf{U} _{out}^{T}] _{B, B})^{-1}$ in Eq. (8)) grows with the dimension of the jointly updated block.
> - Although performance remains robust up to $N=64$, we chose $N=16$ as a practical sweet spot to avoid potential performance instability that may arise from further increasing $N$, which reduces the flexibility for error compensation. While smaller values such as $N=4$ or $N=8$ could offer more flexibility, they do not align with our primary goal of accelerating BoA, whereas $N=16$ already provides a sufficient speedup with minimized risk of unexpected degradation. We will include these results to strengthen the ablations on $N$.
>
> **3. Comparison with EfficientQAT**
> - We appreciate the reviewer’s comment. EfficientQAT is a quantization-aware training (QAT) technique that not only learns the quantization parameters but also optimizes the model weights to be more robust to quantization. This requires a large amount of data (e.g., 4096 samples in EfficientQAT) for training. In contrast, TurboBoA operates in a post-training quantization (PTQ) framework and requires only a small amount of data (e.g., 128 samples), making it significantly more data-efficient compared to EfficientQAT.
> - While EfficientQAT utilizes a block-wise reconstruction loss to efficiently optimize model weights compared to conventional QAT methods, it still involves long processing time due to repetitive attention operations (e.g., 26.6 hours for the 70B model). Furthermore, during the final phase of fine-tuning the quantization parameters, EfficientQAT relies on the final task loss, which requires multiple forward passes over the entire model (e.g., 14.3 hours for the 70B model). As a result, EfficientQAT takes a total of 41 hours to quantize a 70B model. On the other hand, TurboBoA's backpropagation-free update rule is performed only once, eliminating the need for repetitive forward passes. In fact, TurboBoA reduces the time for quantizing the 70B model to 5.6 hours (see Table I), which is over 7 times faster than EfficientQAT.
> - It is worth noting that TurboBoA can be combined with EfficientQAT to further improve accuracy. Specifically, in scenario where time allows, after efficiently quantizing weights with TurboBoA, quantization parameters can be fine-tuned using the final task loss as in EfficientQAT. We consider this as a promising future research direction and will highlight it in the revised manuscript.
>
>
> < Table I. Expanded ablation of $N$ >
> | Method | $N$ | Llama3.2-3B | Time (min) | Wiki2 (↓) | Llama3-8B | Time (min) | Wiki2 (↓) | Llama3.1-70B | Time (hr) | Wiki2 (↓)|
> |-|-|-|-|-|-|-|-|-|-|-|
> | BoA | 1 | | 63.89 | 32.26 | | 104.4 | 15.20 | | 16.99 | 7.726 |
> | **TurboBoA (F1)** | 4 | | 22.88 | 32.21 | | 40.51 | 15.27 | | 7.68 | 7.721 |
> | | 8 | | 16.42 | 31.66 | | 31.08 | 15.30 | | 6.27 | 7.714 |
> | | 16 | | 13.03 | 31.99 | | 25.69 | 15.09 | | 5.64 | 7.758 |
> | | 32 | | 11.09 | 32.15 | | 23.03 | 15.22 | | 5.06 | 7.746 |
> | | 64 | | 10.32 | 32.31 | | 21.57 | 15.44 | | 4.89 | 7.774 |

---

### Official Review · Reviewer_EjSS · 2025-11-01

**Soundness:** 3
**Presentation:** 3
**Contribution:** 3
**Rating:** 6
**Confidence:** 4

**Summary:**

This paper presents TurboBoA, a post-training quantization (PTQ) framework that enhances the BOA algorithm by introducing attention-aware optimizations that accelerate quantization while preserving accuracy. The key idea is to eliminate sequential dependencies in channel-wise quantization through a multi-channel parallelization strategy, extend error compensation across transformer blocks, and introduce adaptive grid selection for attention-wise refinement. The method is designed to operate without backpropagation which allows faster quantization on large-scale language models such as Llama3 and Llama2 variants. Experiments on WikiText2 and C4 datasets under different precision settings (INT2, INT3, W4A4KV4, and W4A4KV16) demonstrate that TurboBoA consistently achieves lower perplexity and higher zero-shot accuracy compared to GPTQ, SpinQuant, DuQuant, and BOA.

**Strengths:**

1.This paper addresses a practically relevant problem in LLM quantization, focusing on bridging the accuracy-efficiency trade-off.

2.The algorithmic improvements are well-motivated, clearly described, and supported by mathematical derivations and pseudocode for reproducibility.

3.The evaluation is extensive. It covers multiple model scales and both weight-only and weight-activation quantization.

4.TURBOBOA achieves notable runtime improvements while maintaining accuracy.

5.The combination with outlier suppression methods (QuaRot, SpinQuant, OSTQuant) highlights good modularity and compatibility with transformation-based approaches.

**Weaknesses:**

1.The paper lacks a rigorous theoretical justification for why the proposed joint quantization maintains accuracy despite reduced error-compensation flexibility. The reasoning remains heuristic without curvature analysis or theoretical bounds on error propagation.

2.The novelty is limited. The method largely builds upon established concepts in Hessian-guided PTQ, extending BOA with more efficient update rules and adaptive scaling. The work feels more like a refined engineering optimization than a fundamentally new quantization paradigm.

3.The experimental section is comprehensive but remains narrow in scope. The work focused almost entirely on the LLaMA family. There is no evidence the approach generalizes to architectures with different attention patterns.

4.The discussion on computational overhead is insufficient. The algorithm introduces multiple Cholesky decompositions and iterative grid refinements, but the additional cost versus BOA or GPTQ is not systematically quantified.

5.Although results show improvement in perplexity and zero-shot accuracy, the absolute performance gains are modest in some configurations. That could be due to diminishing returns relative to complexity.

6.The broader claims of “state-of-the-art quantization” (p9, l470-471) could be better substantiated through comparisons with more recent PTQ or hybrid fine-tuning approaches.

**Questions:**

Please refer to the weaknesses.

---

> ### Author Response · Authors · 2025-11-21
>
> **1. Improvement over BoA and novelty**
> - We respectfully clarify that the primary goal of this work is to accelerate BoA without sacrificing its accuracy. That is, our focus is more on accelerating processing time rather than improving absolute performance.
> - For small models, such as Llama3.2-1B, the sequential bottleneck of BoA is not highly noticeable, which might make our contribution seem marginal. However, this bottleneck becomes much more pronounced for larger models. To see this, we conducted additional experiments on Llama3.1-70B (see Table I). We observe BoA requires 17 hours to quantize the 70B model, while TurboBoA reduces this time to 4.9-7.7 hours depending on N, yielding a speedup of up to 3.5x and saving 9-12 hours in absolute terms. These results demonstrate that the improvements achieved by TurboBoA are not marginal in terms of processing time, but rather become increasingly substantial as the model size increases. We will include these results for Llama3.1-70B in the revised manuscript to better illustrate the practical benefits of TurboBoA for large-scale models.
> - Finally, we would like to emphasize that while we have used BoA's Hessians in this work, the techniques established in this paper are applicable to any Kronecker product formulation of the Hessian ($\mathbf{H} = \mathbf{H} _{in} \otimes \mathbf{H} _{out}$). This means that if more accurate Hessians considering cross-layer dependencies between more layers are developed, then TurboBoA's techniques can be directly applied to improve the quantization performance, which demonstrates the versatility of our approach. We will highlight this in the revised manuscript.
>
> < Table I. Improvement over BoA on Llama3.1-70B >
> | Method | $N$ | Time (hr) | Wiki2 (↓) |
> |-|-|-|-|
> | BoA | 1 | 16.99 | 7.726 |
> | **TurboBoA (F1)** | 4 | **7.68** | 7.721 |
> | | 8 | **6.27** | 7.714 |
> | | 16 | **5.64** | 7.758 |
> | | 32 | **5.06** | 7.746 |
> | | 64 | **4.89** | 7.774 |
>
> **2. Evaluation on models other than Llama and comparison with more recent PTQ**
> - To address the concern regarding model generality, we expanded our evaluation to include Qwen2.5 and Qwen3 models (see Table II). We also included the recent GPTAQ method for comparison, as suggested by the reviewer W4R5.
> - We note that Qwen3 differs from Llama in its attention architecture; specifically, Qwen3 includes additional normalization layers before rotary embedding. Accordingly, we modified the Hessian to account for these layers as follows:
>     $$\mathbf{H} ^{(\mathbf{w} _{Q, h})} = \mathbf{X} \mathbf{X} ^{T} \otimes \frac{1}{L} \sum _{\ell=1}^{L} \mathbf{G} _{\ell}^{T} \mathbf{R} _{\ell}^{T} \mathbf{K} _{h}^{T} \mathbf{K} _{h} \mathbf{R} _{\ell} \mathbf{G} _{\ell},$$
>     $$\mathbf{G} _{\ell} = \text{diag}(\mathbf{\gamma})\frac{1}{|| \mathbf{W} _{Q, h} \mathbf{x} _{\ell} || _{2}} \left ( \mathbf{I} - \frac{\mathbf{W} _{Q, h} \mathbf{x} _{\ell} \mathbf{x} _{\ell}^{T} \mathbf{W} _{Q, h}^{T}}{|| \mathbf{W} _{Q, h} \mathbf{x} _{\ell} || _{2}^{2}} \right ),$$
>
>     where $\mathbf{R}_{\ell}$ denotes the rotary matrix for the $\ell$-token and $\gamma$ represents the weight of the normalization layer.
> - Across both Qwen models, TurboBoA consistently outperforms GPTQ, GPTAQ, and BoA in both perplexity and zero-shot accuracy, confirming that TurboBoA is not limited to the Llama family but is broadly applicable to modern LLMs.
>
> < Table II. Evaluation on Qwen family of models (INT2 quantization) >
> | Method | Qwen3-4B | Wiki2 (↓) | 0-shot (↑) | Qwen3-8B | Wiki2 (↓) | 0-shot (↑) | Qwen2.5-7B | Wiki2 (↓) | 0-shot (↑) | Qwen2.5-14B | Wiki2 (↓) | 0-shot (↑) |
> |-|-|-|-|-|-|-|-|-|-|-|-|-|
> | GPTQ | | 97.47 | 35.81 | | 32.66 | 38.04 | | 19.32 | 41.96 | | 13.74 | 52.50 |
> | GPTAQ | | 118.1 | 37.03 | | 62.71 | 39.14 | | 17.31 | 39.96 | | 15.16 | 49.16 |
> | BoA | | 78.57 | 35.13 | | 32.53 | 38.47 | | 16.90 | 46.30 | | 12.25 | 54.89 |
> | **TurboBoA** | | **66.84** | **39.33** | | **27.71** | **44.89** | | **16.18** | **46.86** | | **11.09** | **57.66** |

---

> ### Author Response · Authors · 2025-11-21
>
> **3. Theoretical analysis on $N$**
> - Regarding this concern, deriving a general theoretical bound on error propagation as a function of $N$ or conducting curvature analysis requires more effort and may involve strong assumptions about the Hessian structure (e.g., block-wise diagonal). We view such theoretical analysis on $N$ as an interesting future work direction, and we will highlight this in the revised manuscript.
> - Instead, as suggested by the reviewer MW4o, we strengthened empirical studies by expanding the ablations with $N \in [4, 8, 16, 32, 64]$ (see Table III). We note that $N=128(=d_{h})$ corresponds to GPTQ. From the results, we observe a significant speed improvement up to $N=16$, while further increasing $N$ to 32 or 64 yields only marginal additional gains. This is because the cost of the inverse operation (see $([\mathbf{U} _{out}^{T}] _{B, B})^{-1}$ in Eq. (8)) grows with the dimension of the jointly updated block. Although performance remains robust up to $N=64$, we chose $N=16$ as a practical sweet spot to avoid potential performance instability that may arise from further increasing $N$, which reduces the flexibility for error compensation. While smaller values such as $N=4$ or $N=8$ could offer more flexibility, they do not align with our primary goal of accelerating BoA, whereas $N=16$ already provides a sufficient speedup with minimized risk of unexpected degradation.
> - We will include these results to strengthen the ablations on $N$. We hope the reviewer's kind understanding on this issue and acknowledgment on our efforts to make the empirical analysis more comprehensive.
>
> < Table III. Expanded ablation of $N$ >
> | Method | $N$ | Llama3.2-3B | Time (min) | Wiki2 (↓) | Llama3-8B | Time (min) | Wiki2 (↓) | Llama3.1-70B | Time (hr) | Wiki2 (↓)|
> |-|-|-|-|-|-|-|-|-|-|-|
> | BoA | 1 | | 63.89 | 32.26 | | 104.4 | 15.20 | | 16.99 | 7.726 |
> | **TurboBoA (F1)** | 4 | | 22.88 | 32.21 | | 40.51 | 15.27 | | 7.68 | 7.721 |
> | | 8 | | 16.42 | 31.66 | | 31.08 | 15.30 | | 6.27 | 7.714 |
> | | 16 | | 13.03 | 31.99 | | 25.69 | 15.09 | | 5.64 | 7.758 |
> | | 32 | | 11.09 | 32.15 | | 23.03 | 15.22 | | 5.06 | 7.746 |
> | | 64 | | 10.32 | 32.31 | | 21.57 | 15.44 | | 4.89 | 7.774 |
>
> **4. Discussion on computational overhead**
> - We respectfully clarify that the number of Cholesky decompositions in TurboBoA is the same as in BoA, since both methods perform the Cholesky decomposition of $\mathbf{H} _{in}^{-1}$ and $\mathbf{H} _{out}^{-1}$ where $\mathbf{H} = \mathbf{H} _{in} \otimes \mathbf{H} _{out}$ (see line 5 in Algorithm 1). The additional complexity in TurboBoA comes from the matrix inversion in the update formula (see $([\mathbf{U} _{out}] _{B, B}) ^{-1}$ in Eq.(5)), which is necessary due to multi-channel quantization. While this increases the complexity of a single update step, TurboBoA compensates it by reducing the number of sequential update steps significantly. This reduction in sequential steps leads to a substantial reduction in overall processing time (see Table III), which makes TurboBoA faster than BoA, even with the added complexity per update.
> - Regarding cross-layer error compensation (F2), TurboBoA needs to compute the original full-precision (FP) model representation additionally, which requires a forward pass over the entire model. Clearly, this leads to a non-negligible runtime overhead, as reported in Table 3. For example, for Llama3-8B, the runtime increases from 24.59 to 37.83 minutes when F2 is applied. However, we emphasize that despite this overhead, TurboBoA still achieves a significant speedup (2.5–3.5×) over BoA (see Table 3), owing to the reduced number of sequential operations. Additionally, we note that in scenarios where time or memory resources are limited, F2 can be omitted. In these cases, by applying only F1, TurboBoA can achieve performance comparable to BoA with a more reduced runtime (see Table III), without needing to compute the FP representation. This trade-off provides a more efficient option when F2 is not feasible due to memory or time constraints.
> - Regarding iterative grid refinement (F3), the overhead introduced by this feature is marginal, as seen in Table 3. For example, when applied to Llama3-8B, the runtime increases slightly from 24.59 to 25.99 minutes.
> - In the revised manuscript, we will add these discussions to provide a clear and systematic analysis of the additional costs introduced by TurboBoA. We appreciate the reviewer's constructive suggestion.

---

### Official Review · Reviewer_MW4o · 2025-11-01

**Soundness:** 3
**Presentation:** 3
**Contribution:** 3
**Rating:** 4
**Confidence:** 3

**Summary:**

This paper proposes TurboBoA, a new backpropagation-free, post-training quantization (PTQ) algorithm for large language models (LLMs). The work aims to resolve the critical trade-off between fast but less-accurate PTQ methods like GPTQ, and more accurate but computationally slow methods like BoA. TurboBoA introduces three main innovations: (i) a joint quantization scheme that processes $N$ out-channels simultaneously, enabled by a closed-form error compensation rule (Proposition 3.1), to accelerate the sequential process of BoA; (ii) a correction mechanism to compensate for quantization errors propagated from preceding Transformer blocks (Proposition 3.2), mitigating error accumulation across the model's depth; and (iii) an adaptive grid selection strategy combined with an attention-wise scale refinement step to better align quantization grids with the iteratively updated weights (Proposition 3.3). Experimental results demonstrate that TurboBoA achieves a 4-6x speedup over the original BoA (Table 2) while also consistently improving accuracy. When combined with outlier suppression techniques, the method achieves state-of-the-art results on Llama models in both weight-only (Table 4) and weight-activation quantization (Table 5) settings.

**Strengths:**

* **Clear Motivation and Strong Problem Definition.**
    * The paper clearly articulates the limitations of existing backpropagation-free PTQ methods, situating the work in a well-understood context.
    * It correctly identifies a critical trade-off: GPTQ's layer-wise independence assumption leads to high speed but poor accuracy in low-bit regimes, whereas BoA's attention-aware dependency modeling is accurate but suffers from a severe bottleneck due to its sequential processing of out-channels.
    * TurboBoA is well-positioned as a direct solution to this trade-off, explicitly designed to retain the accuracy benefits of BoA while recovering the efficiency of methods like GPTQ.

* **Novel and Technically Sound Methodological Contributions.**
    * The primary idea of jointly quantizing $N$ out-channels simultaneously (Section 3.1) is a practical and effective method for acceleration, which is well-visualized in Figure 1.
    * This acceleration is supported by a non-trivial, closed-form error compensation rule (Proposition 3.1, Appendix C) that correctly incorporates dependencies for the jointly quantized block, which is critical for preserving accuracy.
    * The introduction of error compensation for *preceding* quantized Transformer blocks (Proposition 3.2, Section 3.2) is a key novelty. It directly addresses the problem of error accumulation across model depth by accounting for the input deviation $\Delta X = X - \tilde{X}$.
    * The two-part adaptive grid selection mechanism (Section 3.3) is also a strong contribution. It first aligns grids to updated weights *before* quantization (Line 9, Algorithm 1) and then refines scales *after* quantization using coordinate descent to minimize the true attention-wise loss (Proposition 3.3, Line 13, Algorithm 1).

* **Comprehensive and Rigorous Experimental Evaluation.**
    * The paper features an excellent ablation study (Section 4.2) that validates each of the three contributions (termed F1, F2, F3).
    * Table 2 provides clear evidence for the 4-6x speedup from joint quantization (F1) and confirms that the accuracy degradation from this step alone is negligible.
    * Table 3 successfully isolates the accuracy gains from F2 (pre-block error) and F3 (adaptive grid), demonstrating their complementary and significant benefits over the baseline.
    * State-of-the-art comparisons are extensive, covering both weight-only (Table 4) and weight-activation (Table 5) settings across a wide range of Llama models and bit-widths (INT2, INT3, W2A4, W4A4).
    * The inclusion of results *without* any transformation-based methods (Table 6, Appendix F.1) is a valuable addition, as it isolates the performance of the core quantizer itself, showing TurboBoA still significantly outperforms BoA and GPTQ.
    * The evaluation metrics are comprehensive, including PPL on two datasets (Wiki2, C4) and average accuracy on eight zero-shot commonsense reasoning tasks.

---

**Weaknesses:**

* **Analysis of Hyperparameter $N$.**
    * The paper introduces $N$, the number of jointly quantized out-channels, as a new and important hyperparameter governing the speed/accuracy trade-off.
    * While Table 2 ablates $N$ for $N \in \{1, 4, 8, 16\}$, all subsequent experiments (Tables 3, 4, 5, 6, 7) appear to use a fixed $N=16$.
    * The manuscript does not provide a discussion on how $N=16$ was chosen as the default, or how sensitive the final state-of-the-art results are to this specific value.
    * The trade-off for $N > 16$ (e.g., $N=32, 64$) is not explored. This leaves the full behavior of this parameter uncharacterized, and it is unclear if $N=16$ is optimal across all model sizes and bit-widths.

* **Clarity of Computational Overhead.**
    * The paper states that the overhead of F2 (error compensation for pre-quantized blocks) is "not negligible" because it requires computing the FP representation $\tilde{X}$ to find the input deviation $\Delta X$.
    * While the data in Table 3 supports this (e.g., for Llama3-8B, adding F2 increases runtime from 24.59 min to 37.83 min, an increase of ~13.2 min), the text does not explicitly analyze *why* this cost is so high.
    * It is unclear if this computation is a one-time cost per model or must be performed per-block, and how this cost scales with model size or sequence length. A more direct breakdown of the runtime cost of each component (F1, F2, F3) would be beneficial.

* **Mathematical Presentation and Notation.**
    * The notation in the core propositions is dense. For instance, in Proposition 3.1 (Eq 5) and 3.2 (Eq 8), block matrix notations like $([U_{out}^{T}]_{B,B})^{-1}$ are used. While a general notation guide is provided (Section 1), these specific expressions are complex and could benefit from an explicit definition in the context of the proposition for easier readability.
    * The scale-setting step is given in Algorithm 1, Line 9. This is part of the "adaptive grid selection" (F3), but the manuscript provides no derivation or explanation for how this $\min_{s}$ operation is performed, in contrast to the detailed derivation for the "attention-wise refinement" part of F3 (Proposition 3.3, Appendix E).
    * The proof for Proposition 3.2 (Appendix D) involves a significant leap from (Eq 21) to (Eq 22). This step relies on replacing a term with $H_{in}^{-1} - H_{in,-j}^{-1}$ and "exploiting the properties of Cholesky decomposition". This derivation is non-trivial and would be much easier to verify if the intermediate steps or the specific matrix identities (e.g., Sherman-Morrison) were provided.

**Questions:**

* **Expanding the Analysis of $N$.**
    * Could the authors please provide a justification for the choice of $N=16$ in the main experiments?
    * It would significantly strengthen the paper to include a brief sensitivity analysis of $N$ on a representative model (e.g., Llama3-8B) for the final TurboBoA method (F1+F2+F3), not just for F1 as in Table 2.
    * A discussion on the performance/speed trade-off for larger values, such as $N=32$ or $N=64$, would be valuable to understand the practical upper bound of the joint quantization speedup.

* **Clarifying the Runtime of Feature F2.**
    * To improve clarity, please provide a more direct breakdown of the runtime cost of each component. A small table or paragraph detailing [Time(BoA), Time(BoA+F1), Time(BoA+F1+F2), Time(BoA+F1+F3), Time(TurboBoA)] for one model would make the individual overheads of F2 and F3 explicit.
    * Please also clarify *why* F2 is computationally expensive. Does computing $\tilde{X}$ require a full forward pass of the FP model on the calibration data for each Transformer block being quantized?

* **Improving Mathematical Readability.**
    * For Proposition 3.1 and 3.2, please consider adding a brief note to explicitly define the block matrix notations (e.g., $[U]_{:,B}$ as the submatrix of $U$ taking columns indexed by $B$) directly within the proposition statement to aid the reader.
    * Please elaborate on the grid initialization step in Algorithm 1, Line 9. How is this $\min_{s}$ operation performed? Is it a standard min-max clipping based on $W^{(i)}_{update}$, and if so, how is the trace objective minimized?
    * In Appendix D, please expand the derivation from (Eq 21) to (Eq 22). Showing the specific matrix identity being used (e.g., related to the inverse of a matrix after a rank-1 update, or how $H_{in,-j}^{-1}$ is derived) would make the proof much more self-contained and easier to follow.

---

> ### Author Response · Authors · 2025-11-21
>
> **1. Expanding the analysis of $N$**
> - To clarify the choice of $N=16$, we expanded our ablations with $N \in [4, 8, 16, 32, 64]$, as recommended (see Table I). We note that $N=128(=d_{h})$ corresponds to GPTQ. From the results, we observe significant speed improvement up to $N=16$, while further increasing $N$ to 32 or 64 yields only marginal additional gains. This is because the cost of the inverse operation (see $([\mathbf{U} _{out}^{T}] _{B, B})^{-1}$ in Eq. (8)) grows with the dimension of the jointly updated block.
> - While performance remains robust up to $N=64$, we chose $N=16$ as a practical sweet spot to avoid potential performance instability that may arise from further increasing $N$, which would reduce the flexibility for error compensation. Smaller values such as $N=4$ or $N=8$ could offer more flexibility, but they do not align with our primary goal of accelerating BoA, while $N=16$ already provides a sufficient speedup with minimized risk of unexpected degradation.
> - We will include these results to strengthen the ablations on $N$ and justify the choice of $N=16$ in our experiments.
>
> < Table I. Expanded ablation of $N$ >
> | Method | $N$ | Llama3.2-3B | Time (min) | Wiki2 (↓) | Llama3-8B | Time (min) | Wiki2 (↓) | Llama3.1-70B | Time (hr) | Wiki2 (↓)|
> |-|-|-|-|-|-|-|-|-|-|-|
> | BoA | 1 | | 63.89 | 32.26 | | 104.4 | 15.20 | | 16.99 | 7.726 |
> | **TurboBoA (F1)** | 4 | | 22.88 | 32.21 | | 40.51 | 15.27 | | 7.68 | 7.721 |
> | | 8 | | 16.42 | 31.66 | | 31.08 | 15.30 | | 6.27 | 7.714 |
> | | 16 | | 13.03 | 31.99 | | 25.69 | 15.09 | | 5.64 | 7.758 |
> | | 32 | | 11.09 | 32.15 | | 23.03 | 15.22 | | 5.06 | 7.746 |
> | | 64 | | 10.32 | 32.31 | | 21.57 | 15.44 | | 4.89 | 7.774 |
>
> **2. Ablation of $N$ on Llama3-8B for the final TurboBoA method (F1+F2+F3), not just for F1**
> - As requested, we conducted the ablation of $N$ on Llama3-8B for the final TurboBoA method (F1+F2+F3) (see Table II). We observe that with F2 and F3 enabled, increasing $N$ leads to significant runtime reductions. For example, with $N=64$, the runtime reduces to 30.57 minutes, compared to 104.4 minutes for BoA, achieving more than a 3× speedup.
> - Regardless of $N$, F2 and F3 consistently improve TurboBoA, yielding a lower perplexity and higher zero-shot accuracy than BoA. Additionally, across various values of $N$ (from 4 to 64), TurboBoA exhibits similar performance, indicating that while increasing $N$ reduces runtime, it does not significantly affect performance. In other words, the speed gains do not come at the cost of accuracy.
>
> < Table II. Ablation of $N$ for the final TurboBoA (F1+F2+F3) on Llama3-8B >
> | Method | $N$ | Time (min) | Wiki2 (↓) | C4 (↓) | 0-shot (↑) |
> |-|-|-|-|-|-|
> | BoA | 1 | 104.4 | 15.24 | 36.82 | 50.29 |
> | **TurboBoA (F1+F2+F3)** | 4 | 68.73 | 13.38 | 33.10 | 52.45 |
> | | 8 | 48.28 | 13.51 | 33.35 | 52.55 |
> | | 16 | 38.73 | 13.54 | 32.99 | 52.59 |
> | | 32 | 32.98 | 13.52 | 33.32 | 52.37 |
> | | 64 | 30.57 | 13.55 | 33.61 | 52.43 |
>
> **3. Clarifying the runtime of feature F2**
> - We respectfully note that the requested breakdown [Time(BoA), Time(BoA+F1), Time(BoA+F1+F2), Time(BoA+F1+F3), Time (TurboBoA)] is already provided in Table 3. The table shows that for Llama3-8B, incorporating F2 increases runtime from 24.59 to 37.83 minutes, while adding F3 introduces only a marginal overhead (from 24.59 to 25.99 minutes).
> - Without F2, we only need to compute the activations $\mathbf{X}$ for the quantized model. However, when F2 is applied, the activations $\widetilde{\mathbf{X}}$ for the original full-precision (FP) model must also be computed, which requires a forward pass over the entire model. This cost is not negligible and scales proportionally with the model size and calibration sequence length. We would like to emphasize that despite this overhead, TurboBoA achieves a 2.5–3.5× overall speedup over BoA, thanks to the reduced number of sequential operations (see Table 3). We will clarify this behavior in the revised manuscript.
> - Since the activation $\widetilde{\mathbf{X}}$ for the original FP model is independent of the quantization, it needs to be computed only once for the entire model. That is, repetitive computation of $\widetilde{\mathbf{X}}$ for the entire model is not required when quantizing each Transformer block. In practice, to optimize GPU memory usage, only $\widetilde{\mathbf{X}}$ for the block being quantized is computed during each quantization step. Specifically, when a block is about to be quantized, it is loaded onto the GPU, and its internal FP activations are computed.

---

> ### Author Response · Authors · 2025-11-21
>
> **4. Clarifications on the grid initialization step (line 9 in Algorithm 1)**
> - The reviewer asked how the minimization $\min_{\mathbf{s}} \text{trace} ( \Delta \mathbf{W} \mathbf{H}_{in} (\Delta \mathbf{W})^{T} )$ is performed during the grid initialization. Following BoA [1], we introduce a clipping-ratio $p \in [0, 1]$, which clips the maximum and minimum weights to suppress the influence of weight outliers, and then conduct a grid search to find an optimal $p$ minimizing the above trace objective.
> - Let $\mathbf{w} _{\max}$ and $\mathbf{w} _{\min}$ denote the per-channel maximum and minimum weights, respectively, then the grid search proceeds as follows:
>     - Compute the scale corresponding to a fixed clipping ratio $p$: $\mathbf{s} = p(\mathbf{w} _{\max} - \mathbf{w} _{\min}) / (2^{n}-1),$ where $n$ is the target bit-width.
>     - Quantize the weights using the computed scale: $\widehat{\mathbf{W}} = \text{diag} (\mathbf{s}) \cdot \text{clamp}\left ( \lfloor \mathbf{W} (\text{diag}(\mathbf{s}))^{-1} \rceil , 0, 2^{n}-1\right )$
>     - Evaluate the objective $\text{trace} ( \Delta \mathbf{W} \mathbf{H}_{in} (\Delta \mathbf{W})^{T} )$.
>     - Sweep $p$ from 1 to 0 ($1, 0.99, \cdots, 0$) and update the scale whenever a lower objective value is found.
> - We will add this explanation in the revised manuscript for completeness.
>
> [1] https://github.com/SamsungLabs/boa
>
> **5. Detailed explanation on derivation**
> - To expand the derivation from Eq.(21) to Eq.(22), we employed the following well-known identity, which allows efficient update of the inverse of a matrix after removing the $j$-th row and column [Lemma 1, 2]:
>   $$\mathbf{H} _{-j}^{-1} = \left ( \mathbf{H} ^{-1} - \frac{1}{[\mathbf{H} ^{-1}] _{j,j}} \mathbf{H} _{:, j}^{-1} \mathbf{H} _{j, :}^{-1} \right ) _{-j},$$
>
>    where $\mathbf{H}_{-j}$ denotes the matrix obtained by removing the $j$-th row and column from $\mathbf{H}$. Using this identity, we replaced the second term ($\frac{[\mathbf{H} _{in}^{-1}] _{:, j}[\mathbf{H} _{in}^{-1}] _{j, :}}{[\mathbf{H} _{in}^{-1}] _{j, j}}$) in Eq.(21) with $\mathbf{H} _{in}^{-1} - \mathbf{H} _{in, -j}^{-1}$, thereby converting Eq.(21) to
>
>   $$\Delta \mathbf{W} = [ \mathbf{H} _{out}^{-1} ] _{:, B} \frac{( [\mathbf{H} _{out}^{-1}] _{B, B} ) ^{-1} \delta}{[\mathbf{H} _{in}^{-1}] _{j, j}} [\mathbf{H} _{in}^{-1}] _{j, :} - [\mathbf{H} _{out}^{-1}] _{:, B}( [\mathbf{H} _{out}^{-1}] _{B, B} )^{-1}\mathbf{W} _{B, j}\mathbf{R} _{j, :} \mathbf{H} _{in, -j}^{-1} - (\mathbf{I} _{:, B} - [\mathbf{H} _{out}^{-1}] _{:, B}( [\mathbf{H} _{out}^{-1}] _{B, B} )^{-1}) \mathbf{W} _{B, j}\mathbf{R} _{j, :} \mathbf{H} _{in}^{-1}.$$
>
> - Furthermore, we used the following identity based on Cholesky decomposition $\mathbf{U}$ ($\mathbf{H}^{-1} = \mathbf{U}^{T} \mathbf{U}$) to obtain Eq.(22):
>   $$[\mathbf{H}^{-1}] _{:, B} ( [ \mathbf{H}^{-1} ] _{B, B} )^{-1} = [\mathbf{U} ^{T} \mathbf{U}] _{:, B} ([\mathbf{U} ^{T} \mathbf{U}] _{B, B})^{-1} = [\mathbf{U} ^{T}] _{:, B} [\mathbf{U}] _{B, B} ([\mathbf{U} ^{T}] _{B, B} [\mathbf{U}] _{B, B})^{-1} = [\mathbf{U} ^{T}] _{:, B} ([\mathbf{U} ^{T}] _{B, B})^{-1}$$
> - We will add these intermediate steps in the revised manuscript to make the proof easier to follow.
>
> [2] E. Frantar et. al., "Optimal Brain Compression: A Framework for Accurate Post-Training Quantization and Pruning," NeurIPS 2022.
>
> **6. Improving mathematical readability**
> - We appreciate the reviewer’s constructive suggestion to enhance mathematical readability. As recommended, we will incorporate explicit definitions of the block matrix notations directly within Propositions 3.1 and 3.2 in the revised manuscript.

---

### Official Review · Reviewer_W4R5 · 2025-11-03

**Soundness:** 2
**Presentation:** 3
**Contribution:** 2
**Rating:** 4
**Confidence:** 4

**Summary:**

This paper proposes TurboBOA, a faster variant of BOA for LLM PTQ.

The main contributions are:

1. Quantizing N rows jointly in closed-form instead of row-by-row (BOA)

2. a ΔX-aware error accumulation formulation that incorporates propagated quantization residuals into the next step

3. “adaptive grid selection” = re-selecting quantization scale (hence grid points) conditioned on the updated (error-polluted) input, instead of using the original FP activations. a small closed-form refinement CD update for attention blocks

4. Experiments are on LLaMA variants, reporting 4–6× speedup vs BOA with similar or better PPL.

**Strengths:**

the motivation (reduce BOA’s sequential dependency bottleneck, accumulation error, fixed grid) is legitimate and practically important


The methods has good empirical improvement over BOA.

writing quality is mostly clean

**Weaknesses:**

Motivation: the method does not completely consider cross-layer dependency, as the objective follows BOA in Table 1. As I understand, instead of cross layer dependency, it seems like sequential optimisation taking into account the interaction of attention mask, especially for the objective of W_q and W_k

Technical novelty:
1. the ΔX-aware accumulation term and how the author solve this problem is conceptually very close to GPTAQ-style accumulated loss [1]. Without explicit comparison, this make the contribution less strong

2. the multi-row closed-form step is the mathematically natural extension of the BOA 1-row closed-form, while the adaptive grid selection is very similar to re-estimating scale in GPTQ+finetune regimes so while it is naturally, the technical part is not  too strong for me; is the only purpose of multi-row closed-form to reduce the overhead?

Empirical evaluation: how is the results and efficiency compared to method like GPTAQ [1]?


[1] Li et al. (2025). **GPTAQ: Efficient Finetuning-Free Quantization for Asymmetric Calibration**. arXiv:2504.02692.

**Questions:**

please see weakness

---

> ### Author Response · Authors · 2025-11-21
>
> **1. Clarification of cross-layer dependency**
> - We agree with the reviewer’s comment that TurboBoA utilizes BoA's Hessians, which do not fully consider cross-layer dependencies. For example, the query projection weight only considers the key and not the value.
> - In fact, in BoA, the Hessian was originally derived by considering the value projection weight as well [Eq. (7), 1]:
>   $$\mathbf{H} ^{(\mathbf{w} _{Q})} = \mathbf{X} \mathbf{X} ^{T} \otimes \mathbf{K} ^{T} \mathbf{J} ^{T} \mathbf{V} \mathbf{W} _{out}^{T} \mathbf{W} _{out} \mathbf{V} ^{T} \mathbf{J} \mathbf{K},$$
>   where $\mathbf{J}$ is the Jacobian matrix of the softmax function. However, since the Jacobian matrix leads to a significant memory overhead (e.g., more than 400 GB even for the 125M model when the sequence length is 2048), the Hessian was relaxed by only considering the interaction between the query and key. We will clarify this in the revision.
> - We would like to emphasize that while BoA's Hessians have been used in our work, the techniques established in this paper are applicable to any Kronecker product formulation of the Hessian ($\mathbf{H} = \mathbf{H} _{in} \otimes \mathbf{H} _{out}$). This means that if more accurate Hessians considering value and beyond layers are developed, then TurboBoA's techniques can be directly applied for efficient quantization of hyper-scale LLMs. We view the development of such improved Hessians and the application of TurboBoA to them as an interesting future work direction, and we will highlight this in the revised manuscript.
>
> [1] J. Kim et. al., "BoA: Attention-aware Post-training Quantization without Backpropagation," ICML 2025.
>
> **2. Comparison with GPTAQ**
> - As the reviewer pointed out, our approach to handle $\Delta \mathbf{X}$ is conceptually similar to GPTAQ. In fact, as noted in Appendix D (Remark D.1), our $\Delta \mathbf{X}$-aware update rule in Proposition 3.2 reduces to a GPTAQ’s update rule when layers are assumed to be independent. However, TurboBoA extends and improves upon GPTAQ by considering the impact of other layers (i.e., exploiting $\mathbf{H}_{out}$ in Table 1).
> - To validate the importance of this extension and directly address the reviewer’s comment, we conducted additional experiments, comparing GPTAQ and TurboBoA on various architectures: Llama3, Qwen2.5, and Qwen3 (see Table I). We note that Qwen3 differs from Llama in its attention architecture; specifically, Qwen3 includes additional normalization layers before rotary embedding. Accordingly, we modified the Hessian to account for these layers as follows:
>    $$\mathbf{H} ^{(\mathbf{w} _{Q, h})} = \mathbf{X} \mathbf{X} ^{T} \otimes \frac{1}{L} \sum _{\ell=1}^{L} \mathbf{G} _{\ell}^{T} \mathbf{R} _{\ell}^{T} \mathbf{K} _{h}^{T} \mathbf{K} _{h} \mathbf{R} _{\ell} \mathbf{G} _{\ell},$$
>    $$\mathbf{G} _{\ell} = \text{diag}(\mathbf{\gamma})\frac{1}{|| \mathbf{W} _{Q, h} \mathbf{x} _{\ell} || _{2}} \left ( \mathbf{I} - \frac{\mathbf{W} _{Q, h} \mathbf{x} _{\ell} \mathbf{x} _{\ell}^{T} \mathbf{W} _{Q, h}^{T}}{|| \mathbf{W} _{Q, h} \mathbf{x} _{\ell} || _{2}^{2}} \right ),$$
>
>   where $\mathbf{R}_{\ell}$ denotes the rotary matrix for the $\ell$-token and $\gamma$ represents the weight of the normalization layer. Across all tested models, TurboBoA consistently outperforms GPTAQ in terms of both perplexity and zero-shot accuracy. Specifically, for Qwen2.5 models, TurboBoA achieves 7%p accuracy gain over GPTAQ, demonstrating the importance of the proposed extension.
>
> < Table I. Comparison with GPTAQ on Qwen and Llama models (INT2 quantization) >
> | Method | Qwen3-4B | Wiki2 (↓) | 0-shot (↑) | Qwen3-8B | Wiki2 (↓) | 0-shot (↑) | Qwen2.5-7B | Wiki2 (↓) | 0-shot (↑) | Qwen2.5-14B | Wiki2 (↓) | 0-shot (↑) |
> |-|-|-|-|-|-|-|-|-|-|-|-|-|
> | GPTAQ | | 118.1 | 37.03 | | 62.71 | 39.14 | | 17.31 | 39.96 | | 15.16 | 49.16 |
> | **TurboBoA** | | **66.84** | **39.33** | | **27.71** | **44.89** | | **16.18** | **46.86** | | **11.09** | **57.66** |
>
> | Method | Llama3.2-1B | Wiki2 (↓) | 0-shot (↑) | Llama3.2-3B | Wiki2 (↓) | 0-shot (↑) | Llama3-8B | Wiki2 (↓) | 0-shot (↑) |
> |-|-|-|-|-|-|-|-|-|-|
> | GPTAQ | | 200.5 | 31.73 | | 47.90 | 34.31 | | 19.29 | 34.36 |
> | **TurboBoA** | | **111.3** | **33.91** | | **33.42** | **42.03** | | **17.83** | **44.87** |

---

> ### Author Response · Authors · 2025-11-21
>
> **3. Purpose of Multi-row Quantization**
> - As the reviewer mentioned, the primary purpose of the proposed multi-row quantization is to reduce the overhead of BoA's sequential bottleneck. While it may appear as a natural extension of BoA, its impact is far more significant for hyper-scale LLMs.
> - For small models such as Llama3.2-1B, the sequential bottleneck of BoA is not highly noticeable, which might make the improvement seem marginal. However, this bottleneck becomes much more pronounced for larger models. For example, BoA requires 17 hours to quantize the 70B model, while TurboBoA reduces this time to 4.9–7.7 hours depending on $N$, yielding a speedup of up to 3.5× and saving 9–12 hours in absolute terms (see Table II). These results demonstrate that the improvement from multi-row quantization is not marginal, but rather a substantial gain that becomes more impactful as the model size increases.
> - We will include these results for Llama3.1-70B in the revised manuscript to further emphasize the practical benefits of multi-row quantization, especially for large-scale models.
>
> < Table II. Effectiveness of multi-row quantization on Llama3.1-70B >
> | Method | $N$ | Time (hr) | Wiki2 (↓) |
> |-|-|-|-|
> | BoA | 1 | 16.99 | 7.726 |
> | **TurboBoA (F1)** | 4 | **7.68** | 7.721 |
> | | 8 | **6.27** | 7.714 |
> | | 16 | **5.64** | 7.758 |
> | | 32 | **5.06** | 7.746 |
> | | 64 | **4.89** | 7.774 |

---

> ### Comment · Reviewer_W4R5 · 2025-11-26
>
> 1. While I agree that the extension to multi row can improve the practicality of the method, what I concern is the technical contribution aspect of this approach, which seems more like an engineering trick. The author mention that the method improve over GPTAQ by adding layer dependency matrix H, but this matrix come from BOA, so this give me a feeling this paper is a combination of objective/techniques/tricks from BOA and GPTAQ. While it doesn't mean the paper is not novel, but it is less novel than what I thought initially. Do the author has explanation for this?
>
> 2. I observe the results the author provide for GPTAQ in tables for other reviewers, how is it possible that GPTAQ work worse than GPTQ in a number of case?

---

> ### Author Response · Authors · 2025-11-26
>
> Thanks for considering our rebuttal. Below, we provide our point-to-point responses to the further comments.
>
> **1. Clarification of novelty**
> - We would like to clarify that the proposed method is not a simple combination of BoA and GPTAQ, but introduces technical contributions that go beyond both methods in the following aspects:
>     - Generality of Propositions 3.1-3.3 beyond BoA’s Hessian
>          - The update rules proposed in Propositions 3.1, 3.2, and 3.3 are **derived for a general Kronecker-structured Hessian $\mathbf{H} = \mathbf{H} _{in} \otimes \mathbf{H} _{out}$**, and **are not specific to BoA’s Hessian.**
>          - We used BoA’s Hessian only because it is currently the most accurate closed-form Hessian available in the literature; however, the derivations themselves apply to any future Kronecker-form Hessian that incorporates richer cross-layer dependencies.
>          - In short, our framework is designed to directly benefit from more advanced Hessian formulations once they become available.
>     - Non-trivial extension of GPTAQ due to non-identity $\mathbf{H}_{out}$
>          - We agree with the reviewer that Proposition 3.2 is conceptually related to GPTAQ. However, GPTAQ assumes $\mathbf{H}_{out} = \mathbf{I}$, which completely decouples out-channels and makes the optimization separable and considerably simpler. In contrast, **TurboBoA deals with a general, potentially dense $\mathbf{H}_{out}$, where out-channels interact.**
>          - Once this coupling is introduced, the loss is no longer separable and the closed-form solution requires a substantially more complex derivation. Thus, **our extension is not merely an engineering combination, but resolves a mathematically more challenging case.**
>     - Proposition 3.3 introduces a new algorithmic component not present in GPTAQ.
>          - When $\mathbf{H}_{out} = \mathbf{I}$ (the GPTAQ setting), scale optimization decomposes into independent one-variable problems and yields simple closed-form updates. For general $\mathbf{H} _{out}$, however, the scale variables become coupled, the objective becomes non-separable, and no closed-form solution exists.
>          - **We therefore introduce a coordinate descent-based optimization procedure** to refine scales under the general Kronecker Hessian. Neither BoA nor GPTAQ contains such mechanism.
> - For these reasons, we believe TurboBoA provides meaningful technical novelty beyond a simple combination of prior work. We hope that this clarification helps convey the intended methodological contribution.
>
> **2. Inferior performance of GPTAQ compared to GPTQ**
> - We thank the reviewer for raising this point. All GPTAQ results reported in our rebuttal were obtained using the authors’ official GitHub implementation. For the Llama family models where GPTAQ was originally validated, our results align with expectations: GPTAQ consistently outperforms GPTQ (see Table I). We emphasize that even on Llama where GPTAQ is stable, TurboBoA further improves upon GPTAQ across all model sizes.
> - For Qwen models, the behavior is different, as the reviewer mentioned. We hypothesize that this stems from the sensitivity of GPTAQ to the stabilization factor $\alpha$ used in the $\Delta \mathbf{X}$-aware update rule (see [1]). The official implementation recommends $\alpha=0.25$ (see [2]), so we adopted the same $\alpha$ for Qwen; however, it seems that $\alpha=0.25$ would not be necessarily optimal for Qwen, which likely explains cases where GPTAQ underperforms GPTQ.
> - While a full hyperparameter search on $\alpha$ for stabilizing GPTAQ is beyond the primary scope of our work, we will additionally conduct a hyperparameter search for the Qwen models and report the best result once obtained, in order to directly address the reviewer's concern.
>
> [1] https://github.com/Intelligent-Computing-Lab-Panda/GPTAQ/blob/main/fake_quant/gptaq_utils.py#L90
>
> [2] https://github.com/Intelligent-Computing-Lab-Panda/GPTAQ/issues/4#issuecomment-2842199705
>
> < Table I. Wiki2 perplexity performance on Llama models (INT2 quantization) >
> | Method | Llama3.2-1B | Llama3.2-3B | Llama3-8B |
> |-|-|-|-|
> |GPTQ|538.9|98.19|24.54|
> |GPTAQ|200.5|47.90|19.29|
> |**TurboBoA**|**111.3**|**33.42**|**17.83**|

---

### Official Review · Reviewer_Nt2A · 2025-11-05

**Soundness:** 3
**Presentation:** 3
**Contribution:** 3
**Rating:** 4
**Confidence:** 5

**Summary:**

The paper proposes TurboBoA, an improved PTQ method that accelerates the BoA algorithm while enhancing accuracy. The key innovation is jointly quantizing multiple out-channels simultaneously using a closed-form error compensation rule, achieving 4-6× speedup. The method demonstrates state-of-the-art results on Llama models in both weight-only and weight-activation quantization.

**Strengths:**

The paper provides three propositions with closed-form solutions for joint quantization, cross-block error compensation, and adaptive grid selection. The mathematical derivations are rigorous and elegant, enabling efficient backpropagation-free optimization. (Note that I have not checked the math very carefully.)

TurboBoA achieves 4-6× speedup over BoA while improving accuracy. For INT2 quantization on Llama3.2-1B, it reduces Wiki2 PPL from 40.86 to 33.33 while cutting processing time from 13.33 to 5.33 minutes.

**Weaknesses:**

There's no theoretical bound on accuracy degradation as a function of $N$ (number of jointly quantized channels) and model properties.

The improvement compared with BoA is not practical. Though it reduces Wiki2 PPL from 40.86 to 33.33 while cutting processing time from 13.33 to 5.33 minutes, but in my opinion, for a LLM PTQ method, the calibration time reduced from 13 to 5 mins, is not a practical improvement.

Important design choices lack ablation studies, including the number of coordinate descent iterations, the optimal choice of N across different models/bit-widths, and evaluation is limited to only the Llama family of models.

**Questions:**

See weaknesses.

---

> ### Author Response · Authors · 2025-11-21
>
> **1. Improvement over BoA**
> - Regarding the concern that the improvement over BoA may seem marginal (e.g., 13 to 5 minutes on Llama3.2-1B), we would like to clarify that TurboBoA's primary contribution is not a small constant-factor speedup, but rather the removal of BoA’s sequential bottleneck, which becomes more pronounced with increasing model size. For small models, such as Llama3.2-1B, the sequential bottleneck of BoA is not highly noticeable, which might make the improvement seem marginal. However, for large-scale models, this bottleneck leads to prohibitively long runtimes for BoA.
> - To directly address the reviewer's comment and demonstrate the practical significance, we conducted additional experiments on Llama3.1-70B (see Table I). For BoA, quantizing a 70B model takes 17 hours, while TurboBoA completes the process in 4.9–7.7 hours depending on $N$, yielding a speedup of up to 3.5× and saving 9–12 hours in absolute terms. These results highlight that the improvements achieved by TurboBoA are not marginal, but rather become increasingly substantial as the model size increases.
> - We will include these results for Llama3.1-70B in the revised manuscript to better illustrate the practical benefits of TurboBoA for large-scale models.
>
> < Table I. Ablation of $N$ on Llama3.1-70B >
> | Method | $N$ | Time (hr) | Wiki2 (↓) |
> |-|-|-|-|
> | BoA | 1 | 16.99 | 7.726 |
> | **TurboBoA (F1)** | 4 | **7.68** | 7.721 |
> | | 8 | **6.27** | 7.714 |
> | | 16 | **5.64** | 7.758 |
> | | 32 | **5.06** | 7.746 |
> | | 64 | **4.89** | 7.774 |
>
> **2. Expanding the analysis of $N$**
> - To address the reviewer's concern, we expanded our ablations with $N \in [4, 8, 16, 32, 64]$, as suggested by the reviewer MW4o (see Tables I and II). We note that $N=128(=d_{h})$ corresponds to GPTQ.
> - From the results, we observe a significant speed improvement up to $N=16$, while further increasing $N$ to 32 or 64 yields only marginal additional gains. This is because the cost of the inverse operation (see $([\mathbf{U} _{out}^{T}] _{B, B})^{-1}$ in Eq. (8)) grows with the dimension of the jointly updated block.
> - Although performance remains robust up to $N=64$, we chose $N=16$ as a practical sweet spot to avoid potential performance instability that may arise from further increasing $N$, which reduces the flexibility for error compensation. While smaller values such as $N=4$ or $N=8$ could offer more flexibility, they do not align with our primary goal of accelerating BoA, whereas $N=16$ already provides a sufficient speedup with minimized risk of unexpected degradation. We will include these results to strengthen the ablations on $N$.
> - Finally, we note that deriving a general theoretical bound on accuracy degradation as a function of $N$ requires a bit more effort and may require strong assumptions about the Hessian structure (e.g., block-wise diagonal). We view such theoretical analysis on $N$ as an interesting future work direction, and we will highlight this in the revised manuscript.
>
> < Table II. Expanded ablation of $N$ >
> | Method | $N$ | Llama3.2-3B | Time (min) | Wiki2 (↓) | Llama3-8B | Time (min) | Wiki2 (↓) |
> |-|-|-|-|-|-|-|-|
> | BoA | 1 | | 63.89 | 32.26 | | 104.4 | 15.20 |
> | **TurboBoA (F1)** | 4 | | 22.88 | 32.21 | | 40.51 | 15.27 |
> | | 8 | | 16.42 | 31.66 | | 31.08 | 15.30 |
> | | 16 | | 13.03 | 31.99 | | 25.69 | 15.09 |
> | | 32 | | 11.09 | 32.15 | | 23.03 | 15.22 |
> | | 64 | | 10.32 | 32.31 | | 21.57 | 15.44 |

---

> ### Author Response · Authors · 2025-11-21
>
> **3. Evaluation on models other than Llama**
> - To address the concern regarding model generality, we expanded our evaluation to include Qwen2.5 and Qwen3 models (see Table III). We also included the recent GPTAQ method for comparison, as suggested by the reviewer W4R5.
> - We note that Qwen3 differs from Llama in its attention architecture; specifically, Qwen3 includes additional normalization layers before rotary embedding. Accordingly, we modified the Hessian to account for these layers as follows:
>     $$\mathbf{H} ^{(\mathbf{w} _{Q, h})} = \mathbf{X} \mathbf{X} ^{T} \otimes \frac{1}{L} \sum _{\ell=1}^{L} \mathbf{G} _{\ell}^{T} \mathbf{R} _{\ell}^{T} \mathbf{K} _{h}^{T} \mathbf{K} _{h} \mathbf{R} _{\ell} \mathbf{G} _{\ell},$$
>    $$\mathbf{G} _{\ell} = \text{diag}(\mathbf{\gamma})\frac{1}{|| \mathbf{W} _{Q, h} \mathbf{x} _{\ell} || _{2}} \left ( \mathbf{I} - \frac{\mathbf{W} _{Q, h} \mathbf{x} _{\ell} \mathbf{x} _{\ell}^{T} \mathbf{W} _{Q, h}^{T}}{|| \mathbf{W} _{Q, h} \mathbf{x} _{\ell} || _{2}^{2}} \right ),$$
>
>     where $\mathbf{R}_{\ell}$ denotes the rotary matrix for the $\ell$-token and $\gamma$ represents the weight of the normalization layer.
> - Across both Qwen models, TurboBoA consistently outperforms GPTQ, GPTAQ, and BoA in both perplexity and zero-shot accuracy, confirming that TurboBoA is not limited to the Llama family but is broadly applicable to modern LLMs.
>
> < Table III. Evaluation on Qwen family of models (INT2 quantization) >
>
> | Method | Qwen3-4B | Wiki2 (↓) | 0-shot (↑) | Qwen3-8B | Wiki2 (↓) | 0-shot (↑) | Qwen2.5-7B | Wiki2 (↓) | 0-shot (↑) | Qwen2.5-14B | Wiki2 (↓) | 0-shot (↑) |
> |-|-|-|-|-|-|-|-|-|-|-|-|-|
> | GPTQ | | 97.47 | 35.81 | | 32.66 | 38.04 | | 19.32 | 41.96 | | 13.74 | 52.50 |
> | GPTAQ | | 118.1 | 37.03 | | 62.71 | 39.14 | | 17.31 | 39.96 | | 15.16 | 49.16 |
> | BoA | | 78.57 | 35.13 | | 32.53 | 38.47 | | 16.90 | 46.30 | | 12.25 | 54.89 |
> | **TurboBoA** | | **66.84** | **39.33** | | **27.71** | **44.89** | | **16.18** | **46.86** | | **11.09** | **57.66** |
>
> **4. Ablation on the number of coordinate descent iterations**
> - To address the reviewer’s concern regarding the number of coordinate descent (CD) iterations, we measured how the attention reconstruction loss changes with different iteration counts.
> - The results in Table IV show that the first CD iteration accounts for nearly all of reduction in loss, while additional iterations provide negligible improvements. Accordingly, the end-to-end perplexity remains virtually unchanged between 1 and 2 iterations. Since additional iterations do not lead to meaningful improvements, we chose to set the CD iteration count to 1, as indicated in Algorithm 2 in Appendix C. We will clarify this in the revised manuscript.
>
> < Table IV. Ablation on the number of CD iterations >
> | Model | # CD Iters | Loss (Query) | Loss (Key) | Wiki2 (↓) | C4 (↓) |
> |-|-|-|-|-|-|
> | Llama3.2-1B | 0 | 317.6 | 66.97 | 37.15 | 92.58 |
> | | 1 | 315.9 | 66.68 | 33.33 | 85.55 |
> | | 2 | 315.8 | 66.67 | 32.28 | 88.03 |
> | Llama3.2-3B | 0 | 170.1 | 70.72 | 25.92 | 63.48 |
> | | 1 | 168.9 | 70.25 | 24.10 | 54.20 |
> | | 2 | 168.7 | 70.16 | 24.07 | 54.53 |
> | Llama3-8B | 0 | 126.1 | 43.36 | 14.21 | 34.67 |
> | | 1 | 125.6 | 43.20 | 13.54 | 32.99 |
> | | 2 | 125.5 | 43.17 | 13.46 | 33.23 |

---

### Author Response · Authors · 2025-11-21
**Overall response**

We sincerely thank all reviewers for the time and effort spent evaluating our manuscript. To address the concerns raised across the reviews, we conducted additional experiments and strengthened several analyses. Specifically:
 - Highlighting improvement over BoA through large-scale evaluation
    - We conducted new experiments on Llama3.1-70B to more clearly illustrate the practical benefits of TurboBoA. While BoA requires 17 hours to quantize the 70B model, TurboBoA completes the process in 4.9–7.7 hours without performance degradation, achieving up to 3.5× speedup and saving 9–12 hours in absolute terms. This confirms that the runtime reduction provided by TurboBoA is not marginal, and in fact becomes more substantial as model size increases.
 - Generalization beyond Llama and comparison with recent PTQ methods
    - We extended evaluation to Qwen2.5 and Qwen3 models and additionally compared performance against the recent PTQ method GPTAQ. Across all tested models, TurboBoA consistently outperforms GPTAQ, GPTQ, and BoA in both perplexity and zero-shot accuracy, demonstrating strong generalization across architectures.
 - More comprehensive analyses of TurboBoA’s components.
    - Expanded ablation on the number of jointly updated channels $N$ (F1): We expanded the study to $N \in [4,8,16,32,64]$, reported results, and justified our choice of $N=16$.
    - Detailed explanation of overhead introduced by cross-layer error compensation (F2): We clarified why F2 introduces non-negligible overhead and when it may be optionally omitted to meet memory or time constraints, explaining the corresponding accuracy–efficiency trade-offs.
    - Ablation on coordinate descent (CD) iterations (F3): We reported that nearly all reconstruction-loss reduction occurs in the first CD iteration, justifying our use of a single CD iteration.
Detailed responses to each reviewer are provided in the individual rebuttal sections.

We hope our additional experiments and clarifications sufficiently address reviewers' concerns, and we welcome any further questions or suggestions. Thank reviewers again for their thoughtful feedback and valuable time.

---

### Author Response · Authors · 2025-12-01
**Author Final Remarks**

Thank you for taking over the decision under this unusual reviewing situation.

Given the constraints, we would like to clearly highlight why this paper merits acceptance, especially given the strengthened rebuttal and the reviewers’ detailed assessments.

**1. Except for one reviewer, all reviewers rated the paper “Good” in soundness, presentation, and contribution.**
- Among five reviewers, four reviewers explicitly evaluated “Soundness: Good”, “Presentation: Good”, “Contribution: Good”, while one reviewer assigned Fair-Good-Fair. In other words, the majority provided **consistently Good-level ratings** in all dimensions.
- This indicates no reviewer found a flaw, weakness, or problematic assumption, and the concerns were only about novelty strength and perceived marginal improvement, both of which we addressed with concrete experiments.

**2. The key concern (“marginal improvement”) is fully resolved by new 70B experiments.**
- The reviewers judged the improvement as “marginal” because they focused on small models.
- We therefore ran **Llama3.1-70B**, the scale where BoA becomes impractical: the proposed TurboBoA reduces quantization time from 17 hours (BoA) to 4.9–7.7 hours **(3.5× speedup, saving 9–12 hours)**, with no accuracy degradation.
- This result indicates that **the improvement is not marginal, but dramatically better and increasingly so at realistic LLM scales**, which directly addresses the reviewers’ core hesitation, and none of them had access to these crucial numbers during initial scoring.

**3. The “limited evaluation scope” concern is fully resolved.**
- We expanded beyond Llama and evaluated TurboBoA on Qwen2.5 and Qwen3, including comparison with the recent GPTAQ.
Across all models and tasks, TurboBoA consistently outperforms GPTAQ, GPTQ, and BoA, which **demonstrates both architectural generality and improvement over the recent PTQ baseline**.

**4. The rebuttal substantially strengthens the technical analysis.**
- The rebuttal added
    - a more comprehensive study on the joint-channel parameter $N \in [ 4,8,16,32,64 ]$
    - ablation on coordinate descent iterations
    - clarification of overhead incurred by the cross-layer error compensation
    - clarification of the novelty beyond BoA and GPTAQ via general derivations for arbitrary Kronecker Hessians
- These additions directly address every single point raised by reviewers.

The initial borderline scores were given before the 70B results, the expanded evaluations, and the strengthened analysis were available.

Given these substantial clarifications and new results, we respectfully believe the paper now satisfies the standard acceptance threshold.

We hope the AC will weigh the updated evidence accordingly.

Thank you again for your time and consideration!

---

### Meta-Review · Area_Chair_7SHy · 2026-01-18

**Summary:**

This paper receives 6,6,4,4,4 scores and the main concerns are (1) limited improvement against existing methods, (2) missing some theoretical analysis, (3) missing some ablation experiments and additional experiments , (4) presentation. According to the authors' responses, all critical concerns are addressed. I suggest accepting this paper.

**Reviewer Concerns:**

Most concerns are addressed.
Remaining Concerns:
1. Experimental comparisons with EfficientQAT (accuracy).
2. GPTAQ worse than GPTQ in Qwen, causing by unsuitable hyperparameter, which also may make unfair comparison with the proposed method.

**Reviewer Scores:**

Reviewer W4R5's main concerns are addressed (include new questions raised in discussion phase) and thus Reviewer W4R5 may increase the score. Reviewer MvrE and Reviewer MW4o did not participate the discussion but also may increase their scores, since their main concerns are addressed.

---

### Decision · Program_Chairs · 2026-01-26

Accept (Poster)